# A landscape of response to drug combinations in non-small cell lung cancer

Nishanth Ulhas Nair [1], Patricia Greninger [2], Xiaohu Zhang[3], Adam A. Friedman[2], Arnaud Amzallag[2], Eliane Cortez[2], Avinash Das Sahu[4], Joo Sang Lee [5], Anahita Dastur[2], Regina K. Egan[2], Ellen Murchie[2], Michele Ceribelli[3], Giovanna S. Crowther[2], Erin Beck[3], Joseph McClanaghan[2], Carleen Klump-Thomas [3], Jessica L. Boisvert[2], Leah J. Damon[2], Kelli M. Wilson [3], Jeffrey Ho[2], Angela Tam[2], Crystal McKnight[3], Sam Michael[3], Zina Itkin[3], Mathew J. Garnett [6], Jeffrey A. Engelman[2], Daniel A. Haber [2,3], Craig J. Thomas [7,8,9], Eytan Ruppin [1,9] ✉ & Cyril H. Benes [2,9] ✉

Combination of anti-cancer drugs is broadly seen as way to overcome the often-limited efficacy of single agents. The design and testing of combinations are however very challenging. Here we present a uniquely large dataset screening over 5000 targeted agent combinations across 81 non-small cell lung cancer cell lines. Our analysis reveals a profound heterogeneity of response across the tumor models. Notably, combinations very rarely result in a strong gain in efficacy over the range of response observable with single agents. Importantly, gain of activity over single agents is more often seen when co-targeting functionally proximal genes, offering a strategy for designing more efficient combinations. Because combinatorial effect is strongly context specific, tumor specificity should be achievable. The resource provided, together with an additional validation screen sheds light on major challenges and opportunities in building efficacious combinations against cancer and provides an opportunity for training computational models for synergy prediction.

Modern therapeutic approaches to numerous pathologies include the use of drug combinations to obtain better efficacy and lower systemic toxicity in patients. Combinations of drugs have been frequently used to treat microorganisms infections[1,2] and most notably, tri-therapy against HIV infection can yield very long lasting disease control[3]. Drug combinations are also frequently part of anti-cancer treatment, based mainly on empirical clinical discovery for decades[4,5]. Rationally designed targeted agents have now been approved across a variety of cancers but the vast majority of patients are still treated first with combinations of "classic" genotoxic chemotherapeutic agents such as DNA damaging agents or other agents targeting cycling cells (taxanes). Targeted agents are sometimes combined with traditional cytotoxics: e.g., the targeted agent trastuzumab (an antibody against HER2) is combined with taxane to achieve higher benefit in HER2 breast

[1]Cancer Data Science Laboratory, Center for Cancer Research, National Cancer Institute, National Institutes of Health, Bethesda, MD, USA. [2]Massachusetts General Hospital, Harvard Medical School, Boston, MA, USA. [3]Howard Hughes Medical Institute, Bethesda, MD, USA. [4]University of New Mexico, Comprehensive Cancer Center, Albuquerque, NM, USA. [5]Samsung Medical Center, Sungkyunkwan University School of Medicine, Suwon 16419, Republic of Korea. [6]Wellcome Trust Sanger Institute, Wellcome Trust Genome Campus, Cambridge CB10 1SA, UK. [7]Division of Preclinical Innovation, National Center for Advancing Translational Sciences, National Institute of Health, Rockville, MD 20850, USA. [8]Lymphoid Malignancies Branch, Center for Cancer Research, National Cancer Institute, National Institutes of Health, Bethesda, MD 20892, USA. [9]These authors contributed equally: Craig J. Thomas, Eytan Ruppin, Cyril H. Benes. ✉e-mail: eytan.ruppin@nih.gov; cyrilbenes@gmail.com

cancer[6,7]. Currently, there are only few combinations involving exclusively rationally designed targeted agents that are used to treat cancer. There are however notable examples of recent successes: combining CDK4/6 inhibition with Estrogen Receptor (ER) directed therapy is beneficial over other therapies in ER positive breast cancer[8]. In AML, the BCL2 targeting agent venetoclax combined with the demethylating agent aza-cytidine provides substantial improvement in clinical outcome compared to either agent alone or chemotherapeutics regimens[9]. The use of BRAF and MEK1/2 inhibitors in combination has led to improved response in melanoma[10]. Many other targeted combinations are now being tested in clinical trials.

While it stands to reason that combining targeted drugs could improve benefit, the rational development of drug combinations against cancer is still hampered by the limited understanding of underlying cellular processes. There is now ample evidence of heterogeneous response to targeted anti-cancer therapies even within molecularly stratified patients. Indeed, response is still highly variable within the best responsive patient cohorts, with treatment either inefficient up front (innate resistance) or of limited and unpredictable duration (acquired resistance)[11]. Whether combinations of targeted agents will show such heterogeneity in response or allow for more encompassing treatment regimen is not known. Another critical aspect, even for targeted agents, is toxicity. In contrast to drug combinations against HIV for example, targeted drugs against cancer address cellular processes that are almost always shared between cancer cells and normal cells. Consequently, even with targeted agents of good specificity, increased toxicity is a major hurdle for clinical development of combinations and is additionally very difficult to predict. To obtain higher efficacy than single agents and minimize systemic toxicity, drug combinations that are synergistic specifically in cancer cells are thus conceptually attractive even if synergy is not required to obtain effective combinatorial effect on cohorts of patients[12]. Yet, the availability of public large scale combination datasets is limited, additionally impairing efficient computational modeling for combination discovery[13].

In this study we aimed to identify combinations of interest that could help treat non-small cell lung cancer (NSCLC) patients. Through a very large dataset we generated, together with an additional validation screen, we provide a robust estimate of the heterogeneity of response to targeted drug combinations within lung cancers and analyze genetic as well as cellular network determinants of combinatorial effect. This dataset will additionally provide a common grounds resource for the scientific community interested in drug combinations development against cancer, and in the development of computational modeling approaches towards the systematic discovery of synergism in cancer cells.

## Results

### A large-scale drug combination screen in NSCLC models, its design and scoring

To systematically study the response of NSCLC models to pairwise drug combinations, a collection of 81 NSCLC cell lines that are genetically representative of human tumors[14] was assembled. These models are extensively characterized at the molecular level[15]. Mutational profiles for major cancer genes in this collection are shown in Supplementary Fig. 1. Similarly to what is seen in exome sequencing data of human tumors[16] only a handful of cancer genes[17] are recurrently mutated across the cell line collection (Fig. 1A). Recently, fusion events were systematically identified for 79 out of 81 cell lines, most of which identified were not associated with a clear functional role[18], and were thus not studied for their relation to drug combination response here (except for EML4-ALK).

The response of the cell lines used in the present study to single-drug treatments was previously studied comprehensively across >400 single agents[15]. In addition, 49 of the cell lines were also part of a large

chemical screening effort performed across NSCLC lines surveying an initial set of >200 K compounds and an activity based selected subset of 447 chemical entities[19]. These single-agent datasets as well as the results of genetic perturbations using shRNA[20], super potent siRNA pools[21] or more recently CRISPR CAS9 mediated loss of function[22–24] demonstrated that these NSCLC models capture the clinically relevant aspects of therapeutic response of the disease. Importantly, as observed in the clinic, these data also demonstrate a prevalent heterogeneity of response to a given perturbation even within subsets of models sharing a common oncogenic driver (heterogeneity of response within KRAS driven NSCLC models for example ref. 21).

To identify active drug pairs across the 81 cell lines, 21 "anchor" drugs were selected based on their relevance to NSCLC treatment, approval status, results of preclinical therapeutic studies and biology. Those were combined with 242 "library" drugs covering most targeted therapeutic classes currently in use or in development against cancer. This $21 \times 242$ testing strategy was used in an ultra-high throughput screen in 1536 well plates using one fixed dose of anchor drug and five doses for each library drug (Fig. 1B, Supplementary Data S1, S2). Figure 1C lists the anchor drugs used and (Fig. 1D) summarizes the targets and classes of library drugs. The dosing strategy of anchors and library drugs was aimed at discovering combinations with strong effect on viability (determined here using enumeration of nuclei across treatments). For this, drug dosages achieving complete or near complete targets suppression was sought. This strategy has previously been successful in discovering combinations to counter acquired resistance but is not conceptually restricted to this case[25]. The concentrations of the anchor drugs were chosen based on prior knowledge of on-target potency in cells and profile of response of these drugs across several hundreds of cell lines when available from prior studies (GDSC web site). A large-scale single-agent screen data was used to determine the dose of anchor drug that yielded very strong viability suppression in only a few cell lines (typically less than 2% of >500 cell lines tested). For EGFR inhibitors this would correspond to the highly sensitive cell lines that are dependent upon EGFR activity. The underlying assumption is that while the viability outcome of target inhibition varies across cell lines, a given drug will overall affect its target(s) equivalently across cell lines (barring drug pumps effects which in fact do not strongly affect the vast majority of drug responses in cells[15]). Indeed, while the complete lack of target expression is expected to ablate drug effects, previous large scale drug screening studies in cancer cell lines have shown that apart for a few exceptions concerning amplified genes, the RNA expression of the target is not a strong predictor of drug effects[15] or siRNA mediated target depletion[20]. Thus, the anchor doses correspond to near complete suppression of target activity, which was for most targeted drugs ineffective in the majority of cell lines[15]. Similarly, for the library drugs, the concentration yielding strong viability suppression in only a few cell lines was determined based on single agent data or relevant literature. To further ensure that library drugs were suppressing their target(s) efficiently, one higher dose was added above this informed dose. Three additional lower doses were added to survey a larger breadth of target suppression. A dilution scheme of √10 was used (tenfold dilution every other dose). Drugs and concentration used are listed in Supplementary Data S1. The viability distribution for each single library drug and anchor across all doses is shown in Supplementary Fig. 1 demonstrating that the dosing strategy did yield an appropriately broad range of viability across cell lines.

The screen was performed in technical duplicates with two sets of identical plates seeded on a given day: two DMSO anchored plates corresponding to single agent treatments and two anchor plates corresponding to combination treatments. Screening was repeated for

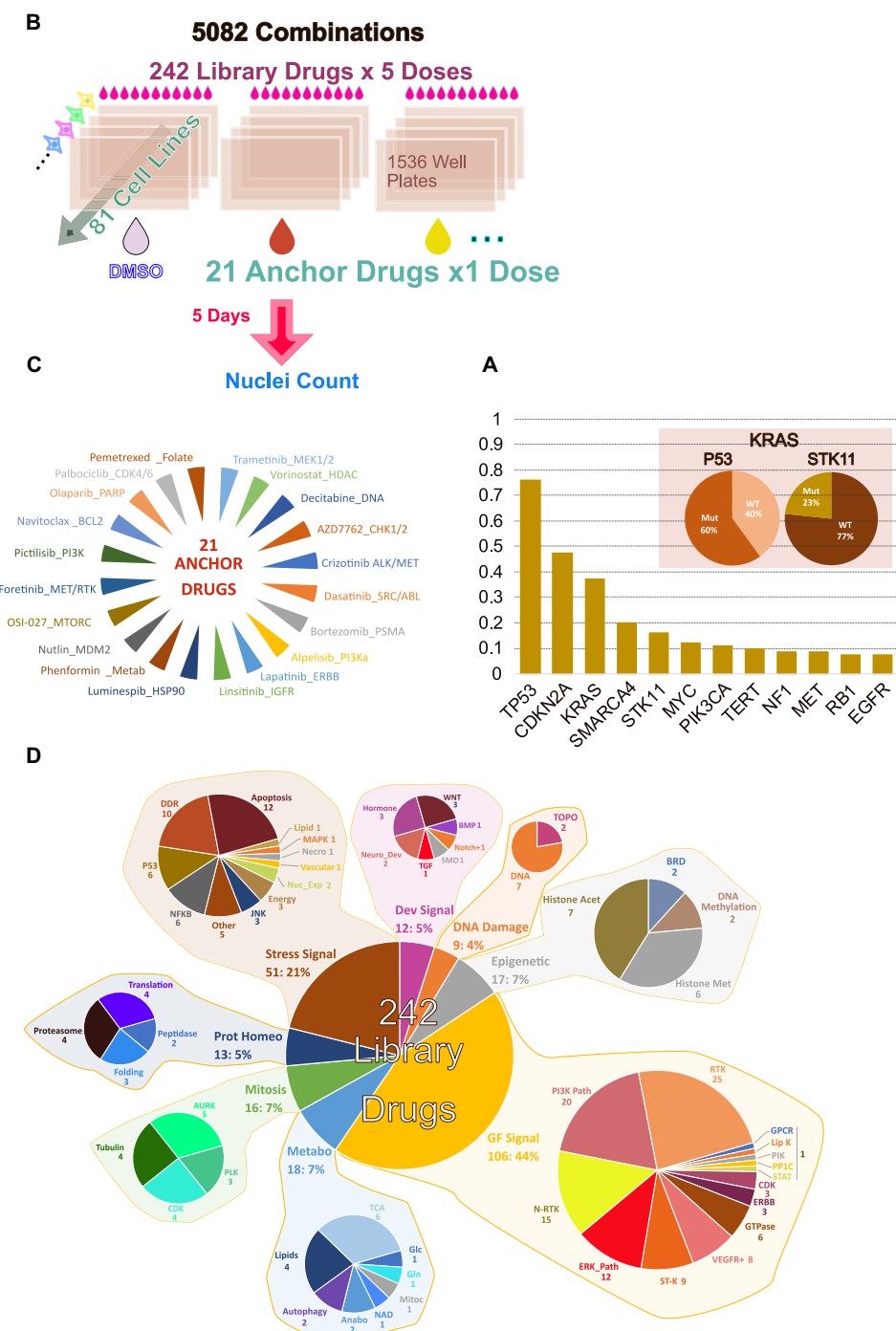

**Fig. 1 | Overview of the study screening strategy. A** Major cancer drivers captured by the cell line collection screened. The y-axis in this figure shows the fraction of cell lines with a driver mutation. Figure also shows the percentage of KRAS mutated cell lines which also have a P53 and STK11 mutation. **B** Screen set-up and key characteristics. **C** Anchor drugs used. **D** Library drugs used grouped by target class. Abbreviations used – DDR DNA Damage Response, Nuc_Exp Nuclear Export, Necro Necroptosis, Lipid Lipid Synthesis, Neuro_Dev neuronal development, Histone Acet Histone Acetylation, Histone Met Histone Methylation, Prot Homeo Protein Homeostasis, Metabo Metabolism, Anabo Anabolism, Mitoc Mitochondria, N-RTK Non Receptor Tyrosine Kinase, RTK Receptor Tyrosine Kinase, Lip K Lipid Kinase, ERK_Path ERK Pathway, ST-K Serine Threonine Kinase.

plates that failed quality control based on coefficient of variation (CV < 25%) of the control wells (DMSO or anchor alone). To collect data on all anchors, a given cell line had to be seeded repeatedly on different days. With the goal of minimizing noise in the dataset, single agent testing (DMSO as anchor) was repeated in parallel with each anchor to allow matched DMSO anchored plates and combination plates of the same drugging run to be compared. Thus, throughout the analyses, combination (anchor plate) and single agent (DMSO plate)

data are compared using only plates matched by cell seeding date. Prescreening calibration of the cell density allowing for proper proliferation and good cellular enumeration was performed. Failure rate varied across cell lines but was overall low: in total, 5766 plates (1536 well plates) were used and 4223 passed QC requiring the cell population to double at least once in addition to acceptable CV of control wells. 73 cell lines out of 81 tested had a pass rate above 90% and only two had a pass rate below 50%. Thus, while a small number of

combinations were not captured in the QC passed dataset for a minority of cell lines, the overall data coverage is high, and the vast majority of tests were performed at least in two technical replicates (Supplementary Fig. 1).

To evaluate the quality of the data the correlation of viability values across technical replicates was computed. There was an overall good correlation across technical replicates with Pearson's *R* value across DMSO plates (single agent library + DMSO) of 0.80 and for technical replicates across combinations plates (Anchor drug + library drug) of 0.76. To evaluate the outcome and overall value of the screen data, a measure of synergy based on statistical independence of effect of the single agents was used (Bliss model): synergy was determined by considering the outcome of each single agent, taking the product of the single agent effects as the predicted outcome and comparing it to the experimentally determined viability outcome of the combination. The ratio between the expected and the observed outcomes constitutes the primary metric of synergy at each tested dose pair and an overall synergy score is derived from these five values (five doses of library drug combined with one dose of anchor drug). A synergy score <1 implies the drug combination is synergistic, with lower values indicating higher synergy (Methods, Supplementary Note 1, Supplementary Data S2). To increase the likelihood of true positives we considered the five-dose pairing individually and extracted the 2nd highest synergy from the series of five values, denoted as the *synergy score* of that combination. This 2nd best from the five synergy values can therefore correspond to any of the doses tested (not necessarily the 2nd maximum dose tested). To complement this synergy score an efficacy gain score was also computed: The Higher than Single Agent (HSA) score describes the additional viability loss observed with a combination over the maximum viability loss observed with either of its components individually. HSA measure does not make any assumption regarding independence of effect of drugs or other assumptions built into various models of synergy[26]. Here, a negative HSA score implies the drug combination is more effective than the better of the two drugs (Methods). To confirm the validity of our drug interaction scores, empirical *p* values for synergy and HSA scores were also computed and a very strong correlation was found between those and the interaction scores (Methods, Supplementary Data S7). Furthermore, the correlation between the second-best and the median synergy values across all cell lines and drug combinations is reassuringly very strong (Spearman's $\rho$ = 0.79, $P < 2.2e-16$). To obtain a ranking of synergistic drug pairs, two complementary strategies were initially used, leveraging the synergy score: (i) computing the median synergy score across all tested cell lines and (ii) the count of cell lines with a synergy score of 0.8 or less (see below how different synergy score impact the number of synergies observed across cell lines). Similarly, for HSA, a global score was obtained by either (i) taking the median of all HSA scores for that combination across cell lines or (ii) counting the number of cell lines passing a threshold of 15% HSA (loss of 15% of cellular viability compared to the viability obtained with most effective of the two single agents).

Using these metrics, a set of combinations known to yield benefit over single agents or straightforwardly mechanistically supported were then scrutinized. For example, let us describe the results obtained with the anchor AZD7762 an inhibitor of the DNA damage repair response (DDR) kinases CHK1 and CHK2. Ranking combinations with AZD7762 based on the number of cell lines where the combination effect is either superior to single agents' (HSA 15% or more) or synergistic (synergy score of 0.8 or less) shows that the inhibitor of WEE1, a kinase that regulates cell cycle checkpoint, is the top combination partner for AZD7762. Multiple synergies were also seen when combining ATR and CHK1/2 inhibitors (Fig. 2). There is published evidence for synergy between CHK1 and WEE1 inhibition[27–29]. The DNA damaging agents cytarabine, gemcitabine as well as the antimetabolites pemetrexed and 5-FU also displayed HSA/Synergies in combination with AZD7762 albeit in fewer cell lines in the later cases than the former (Fig. 2). Thus, there is clear detection of signal for combinations that were expected to be synergistic based on pathway knowledge and previous literature[28,30]. An overview of the screen outcome based on counts of synergy events across cell lines is presented in Supplementary Fig. 2 (see also Supplementary Data S2c).

To systematically identify top combinations for each anchor and estimate how impactful a given combination might be in the clinic, an *impact score* for each drug combination was computed based on the distribution of synergy scores across cell lines for each combination: this impact score was computed by comparing the distribution of synergy scores (or separately HSA scores) across cell lines for a given drug combination with the distribution of scores for all other drugs combined with the same anchor, using a Wilcoxon rank sum test. As a secondary measure, the median of the scores across cell lines for a given drug was compared to the median of scores of the rest of the drugs. The top combinations identified represent those with the highest effect across cell lines and thus perhaps across NSCLC patients. To further characterize the most promising combinations, the percent of cell lines with a synergy score within the top 5% of all scores (all anchors) was also computed (some of the top synergies observed are described in Fig. 3). This systematic approach readily identified the combination of WEE1 inhibitor with CHK1/2 inhibitor and other combinations described above as the most impactful combinations for the CHK1/2 anchor (Fig. 3A). Below we describe the top ranked combinations identified in our screens, based on their impact scores (Fig. 3G shows a summary of the top combinations identified).

## Salient combinations across anchor drugs

Here we review and discuss our results for specific combinations of interest that reaffirm and expand upon previous studies.

We start with combinations involving the parylating enzymes Poly-ADP-Ribose-Polymerases (PARP), probably the most well-known examples of synthetic lethal based treatments to date. Somatic mutations *BRCA1/2* are found across several cancer types and can confer clinical sensitivity to PARP inhibitors[31], such as olaparib. Top HSA partners we identified for olaparib include decitabine and zebularine, two related agents known to induce demethylation of DNA. Decitabine was also used as an anchor in the present study and olaparib was its top ranked synergistic partner with another PARP inhibitor veliparib ranking second (Fig. 3B).

The MEK inhibitor trametinib (first approved by the FDA for use in BRAF V600E melanoma) has been studied in combination across a variety of contexts. Feedback re-activation of the MEK pathway upon suppression of MEK or ERK impairs the clinical activity of BRAF inhibitors[32–34]. Synergistic activity of RAF and MEK inhibitors combination has indeed been documented[35–37]. Here, the pan RAF (A, B, C-RAF) inhibitor AZ628 was the top combination partner for trametinib. The ERK inhibitor VX11e also yielded frequent synergies with trametinib (Fig. 3G). Consistent with published reports on treatment benefit in preclinical models[38–40], synergies were also frequently observed between trametinib and inhibitors of the PI3K/mTOR pathway (Fig. 3D, Supplementary Figs. 2, 5C). Across receptor tyrosine kinase inhibitors, those targeting insulin receptor / insulin growth factor receptor led to more synergies than combination with ERBB family members (Supplementary Fig. 3D).

Drugs targeting the PI3K pathway also yielded interesting outcomes. The PI3K inhibitor alpelisib (BYL719) which targets selectively the alpha catalytic isoform of PI3K (encoded by *PIK3CA*, frequently mutated across multiple cancer types) was tested as an anchor drug. Combining BYL719 with PI3Kbeta selective inhibitors yields a strong HSA and synergistic effects across many cell lines with good consistency seen between the two PI3Kbeta inhibitors tested (AZD6482 and TGX221, Fig. 3E, Supplementary Figs. 3–5), in concordance with earlier reports in

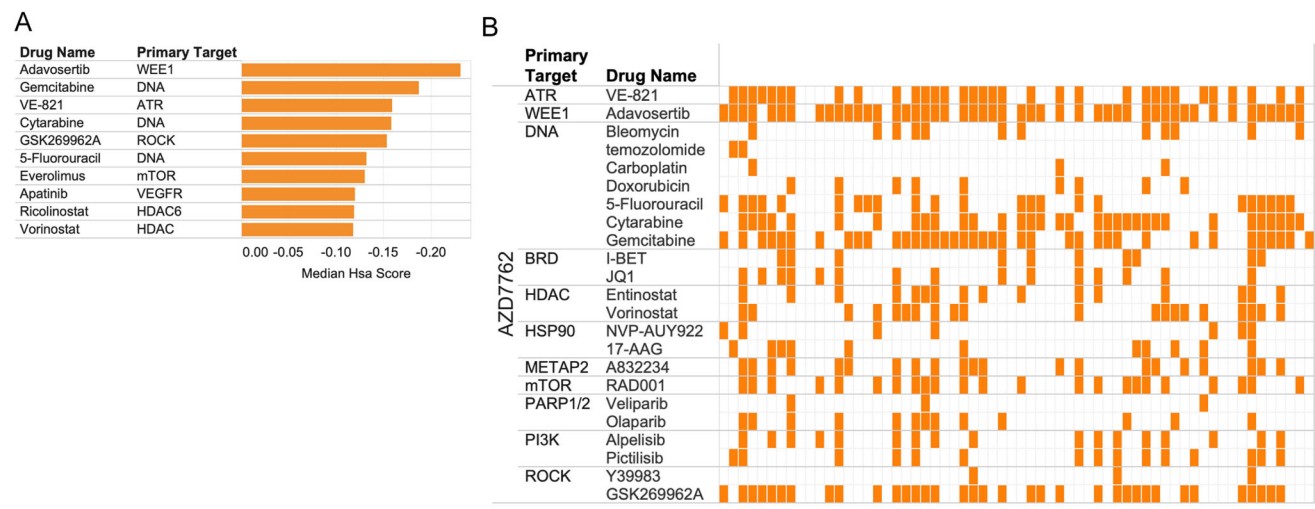

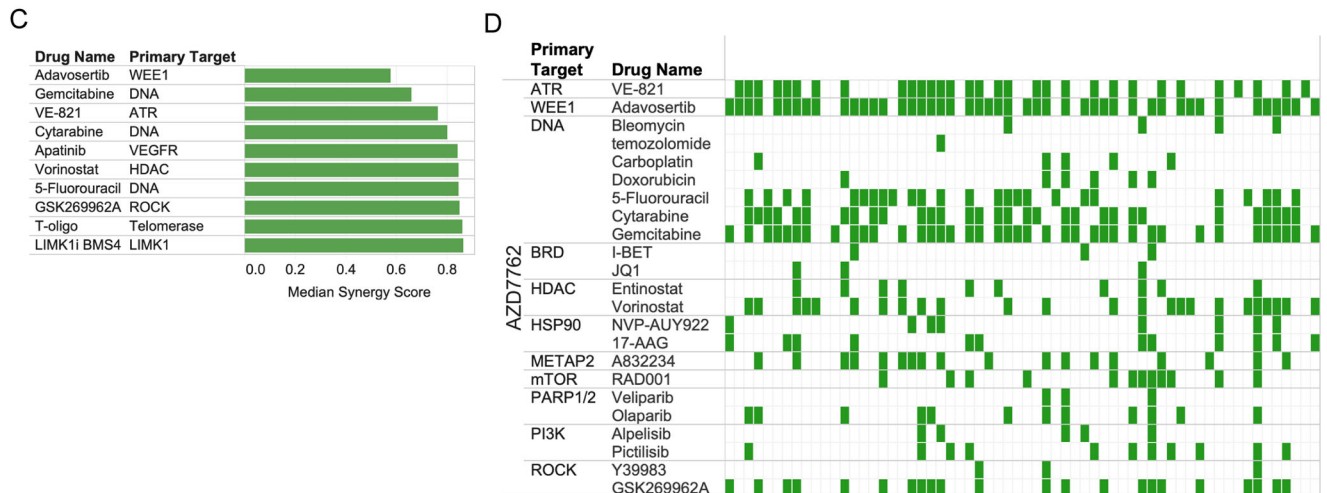

**Fig. 2 | Capture of expected combinatorial effects. A** AZD7762 combinations: top combinations sorted by median score across cell lines using Higher than Single Agent metric. **B** Pattern of HSA events across top combinatorial partners for AZD7762 across cell lines. Each row corresponds to the specified drug combined with AZD7762 and each column (mark) corresponds to a cell line. Colored marks correspond to positive HSA events. The cell lines are in the same order across rows revealing differential pattern of HSA events for different combinations. **C** Top combinations (with AZD7762) based on median synergy across cell lines. **D** Pattern of synergistic events displayed as in (**B**) but using synergy rather than HSA.

breast cancer[41]. EGFR family inhibitors also display relatively frequent HSA with BYL719 across cell lines[42,43]. By contrast, the inhibition of other non-receptor tyrosine kinases of the SYK family or inhibition of FGFRs display no combinatorial benefit with BYL719. The pan-PI3K inhibitor pictilisib (GDC0941) and the MTORC inhibitor OSI-27 were also used as anchors: Pictilisib was broadly synergistic with trametinib, the ERK inhibitor VX11E and the mTOR inhibitor RAD001 (everolimus) (Fig. 3E, Supplementary Figs. 3B, 4, 5C). Similarly, pathway combinations of OSI-27 with PI3K and AKT inhibitors were synergistic in many cell lines, and ERK or MEK inhibitors were also among the top synergizing drugs with OSI027 (Supplementary Figs. 4, 5, 2). Combining a catalytic inhibitor of MTORC1/2 and everolimus was previously shown to yield synergistic inhibition of MTORC1[44] and this was apparent here in the viability outcome. The insulin/insulin growth factor receptors inhibitor BMS754807 was the top RTK inhibitor synergizing with PI3K inhibition. Indeed, the insulin receptor family is a potent and major (even likely the ancestral) activator of PI3K amongst RTKs[45,46].

CDK4/6 inhibition has been recently reported to be synthetic lethal with an array of partners. Here, the FDA approved CDK4/6 inhibitor palbociclib displayed strong synergies that match relatively well the recent data demonstrating the clinical relevance of the interaction between the inhibition of the PI3K/mTOR and CDK4/6 inhibition[47]. MEK and ERK inhibition were also seen as producing some synergies with Palbociclib. Selicicilb (CDKs) and I-BET (BRD) were the top combination in terms of number of synergies. HSA analysis confirmed BRD targeting drugs JQ1 and I-BET as some of the top combinations with palbociclib (Supplementary Figs. 3, 5). The most cell lines with HSA were obtained with inhibition of mTOR and a number of strong HSA scores were seen with trametinib[48,49]. Overall, however, relatively few synergies were seen with palbociclib and consequently their impact scores were low (which is why this anchor is not present in the overview presented in Fig. 3G).

The inhibitors of the mitotic kinases AURK and PLK are among the drugs presenting with the most synergies with vorinostat, the anchor HDAC inhibitor. The proteasome inhibitor carfilzomib, the NEDD8 activating enzyme (NAE, involved in E3 Cullin family activation) inhibitor MLN4924, the BET inhibitors I-BET and JQ1, the LSD1 inhibitor LSD1-C76 and the topo-isomerase I inhibitor irinotecan showed many synergies with vorinostat. These are well supported by literature in preclinical and for some, clinical studies (leukemia, cutaneous T-cell

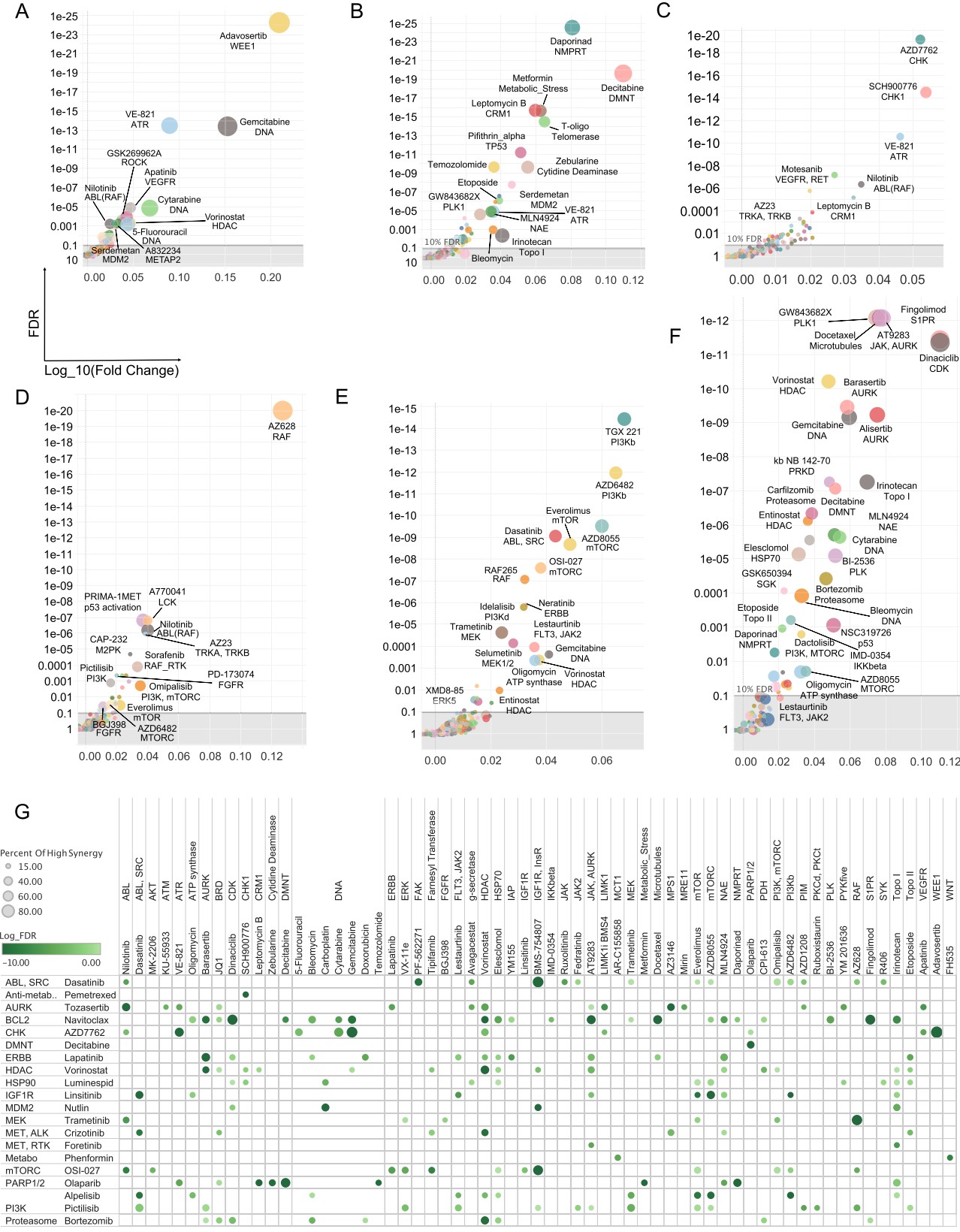

lymphoma for which vorinostat is an approved agent, multiple myeloma[50]). Synergies with all three AURK inhibitors tested are strong and numerous suggesting on-target basis for the observed effects. Synergy between one of the AURK inhibitors tested here, alisertib and the HDAC inhibitor romidepsin was reported in T-Cell lymphoma[51]. Agents targeting metabolic enzymes were also good combinatorial

partners for vorinostat: CPI613 (PDH/aKGH), Bromopyruvate (Hexokinase) (Supplementary Fig. 4E).

Numerous growth factor pathway inhibitors synergize with the tyrosine kinase inhibitor dasatinib, an inhibitor of ABL, SRC family kinases (SFK) and multiple other tyrosine kinases. Multiple synergies were seen across inhibitors of the ERK and PI3K/mTOR pathways

**Fig. 3 | Top synergistic combination across anchors. A–F** The impact score for each combination is plotted with median synergy score of a given combination across cell lines compared to the median synergy score of all other pairs for that anchor represented on the X axis (as the Log10 ratio of median scores) and statistical enrichment of synergies for the plotted combination over all other tested combinations (with the same anchor) represented on the Y axis. The size of the dots represents the percentile of synergy scores for a given combination falling within the top 5% of all synergy scores for the whole screen (all anchors). Each plot corresponds to a different anchor drug: **A** AZD7762 (CHK1/2) **B** Olaparib (PARP) **C** Pemetrexed (anti-folate); **D** trametinib (MEK1/2) **E** alpelisib (PI3Kalpha) **F** navitoclax (BCL2). **G** Overview of the top combinations based on impact score FDR and percentile of events (cell lines presenting with synergy) in the top 5% of all synergy scores (across all anchors: strong synergies). Combinations with at least 15% of strong synergistic events are shown (15% of the synergy scores across cell lines for that drug pair fall in the top 5% synergy scores overall). The size of the dots corresponds to percentile of events in the top 5% and color shade to the statistical enrichment (FDR).

(Supplementary Figs. 3–5). Consistent synergies were seen across EGFR family inhibitors and combination with INSR/IGF1R inhibitors yielded numerous HSA (Supplementary Fig. 4). The FAK inhibitor PF-562271 synergized strongly with dasatinib across multiple cell lines, the JAK inhibitors TG101348 and ruxolitinib both synergize frequently with dasatinib albeit at a low level. Some synergies with the BTK family inhibitor ibrutinib (PCI32765, FDA approved for use against several hematological cancers which also inhibits BMX, a BTK family member expressed in carcinoma[52] are also seen. All of these are in keeping with known signaling interactions between SFKs and FAK, BTK, JAK, and RTK family members[53]. Overall, dasatinib, perhaps due to its high level of polypharmacology is broadly synergistic but with top synergistic partners in keeping with known roles of SFKs in signal transduction.

A variety of other synergistic combinations emerged in the screen, which, due to space limitations are described in detail in the Supplementary Note 3. Those include the finding that the BCL2 family inhibitor navitoclax is frequently synergistic with cell cycle blockers and strongly synergistic with the approved sphingosine receptor modulator fingolimod (Fig. 3F), and the discovery of many additional synergistic combinations for which no previous report exists as far as we can tell, although in some instances indirect supporting evidence does exist. For example, the CHK1/2 inhibitor AZD7762 synergizes with the ROCK inhibitor GSK269962A (Figs. 2, 3A) and a functional interaction between ROCK and DNA damage repair has been reported[54]. AZD7762 also synergizes with the METAP2 (Methionine Aminopeptidase) Inhibitor, A832234. There is precedent for regulation of cell cycle and METAP targeting[55]. Some of the strongest observed synergies are only found in very few cell lines and are thus not flagged by the impact score analysis presented in (Figs. 2, 3G). Nevertheless, these highly context specific synergies might be mechanistically revealing and could be interesting to explore in other tumor types, where they might be more broadly relevant. The top synergistic combinations (top 5% of all synergy scores across all anchors) are represented as a network of Anchor-Library drugs interactions in Supplementary Fig. 8. Interestingly, there is a high number of drugs that are shared between anchors among the top synergistic pairs. This suggests there may be core dependencies in the NSCLC lineage and some biological processes and regions of the cellular interactome that could be prioritized for further explorations.

Finally, we note that although the present work focused on combinations of targeted agents, pemetrexed, a relatively well tolerated cytotoxic agent and one frequently used to treat NSCLC, was chosen as an anchor given that its administration is frequently associated with emerging resistance. A systematic screen of cytotoxic agents has recently been published across the NCI60 collection of cell lines[56]. Consistent with its mechanism of action and previous studies[57], the top three synergistic drugs with pemetrexed were all inhibitors of the DNA damage response (Fig. 3C). Few other strong synergies were detected across the rest of the library drugs including with motesanib (RTKs) and nilotinib (ABL, RAF,TKs). Interestingly, different cytotoxic agents gave distinct patterns of synergy even with drugs of similar MOA (such as DNA damaging agents). A striking example is the differential synergy profiles of vincristine and docetaxel. Both are targeting microtubules albeit through different mechanisms, but docetaxel displays many more synergies across anchors than vincristine. This doesn't appear to be simply due to poor dosing choice for vincristine as there are instances of anchors displaying more synergies for vincristine than docetaxel (OSI027, dasatinib, phenformin, see Fig. 4D). Overall, these results illustrate that there are likely important drug specific activities that need to be considered to select the most appropriate pairs of drugs (rather than only targets) (Fig. 3G, Supplementary Fig. 6).

## Impact of compound polypharmacology on combination outcome

Analysis of the characteristic properties of many of the synergistic combinations discovered reveals a few key emerging insights and principles, which we describe henceforth.

Because synergism emerges from the functional relation between targets there is considerable complexity to expect when drugs with multiple targets are combined. To study how polypharmacology (engagement of multiple often unrelated targets by a given drug) affects synergy, the synergy patterns of drugs sharing some targets but differing in others were compared. First, we begin with some notable cases. Figure 4A shows the outcome of the comparison between imatinib and nilotinib. Striking differences can be observed with a much larger number of strong synergies observed with nilotinib, which targets RAF in addition to ABL, which is targeted by both. Similarly, comparing four drugs targeting Aurora kinases (AURK) in combination with the HDAC inhibitor vorinostat (Fig. 4B) and four different ERBB family inhibitors combined with the MTORC inhibitor OSI0927 (Fig. 4C) shows that while the number of synergies and strength of those synergies are qualitatively similar, some synergies are unique to specific drugs. Figure 4D plots the synergy profile of library drugs targeting cell cycle entry and progression (see also Supplementary Fig. 3). As expected, there is an overall similarity of their synergistic behavior across the majority of anchors. However, while genetic studies indicate some level of functional redundancy between CDK2,4 and 6, the independent targeting of either CDK2 (dinaciclib) or CDK4/6 (palbociclib, PD-03329921) can yield numerous different synergies. There are striking differences between CDKs inhibitors combinations with dinaciclib[58], showing a much more active profile than seliciclib (roscovitine, CDK1/2/5/9), with only few anchors including OSI-027 (mTORC) and palbociclib (CDK4/6) displaying more synergies with seliciclib than dinaciclib. Because both dinaciclib and seliciclib inhibit CDK1/2/5/9 equipotently (at least in vitro[59]) it appears that secondary target(s) or perhaps differential mode of target engagement[60] might be underlying the differences observed. Notably, we also find that AURK inhibitors had a clear tendency to be more broadly synergistic than CDK inhibitors even though both compound classes target kinases best understood as cell cycle regulators (Supplementary Fig. 7).

## Validation of selected key findings in an additional screen

The use of 81 cell lines likely allowed us to robustly capture true synergistic events as well as estimate which combinations could be most relevant for disease treatment. To further test the robustness of our results and validate some of the key findings of our original screen, we carried out an independent screen of selected combinations

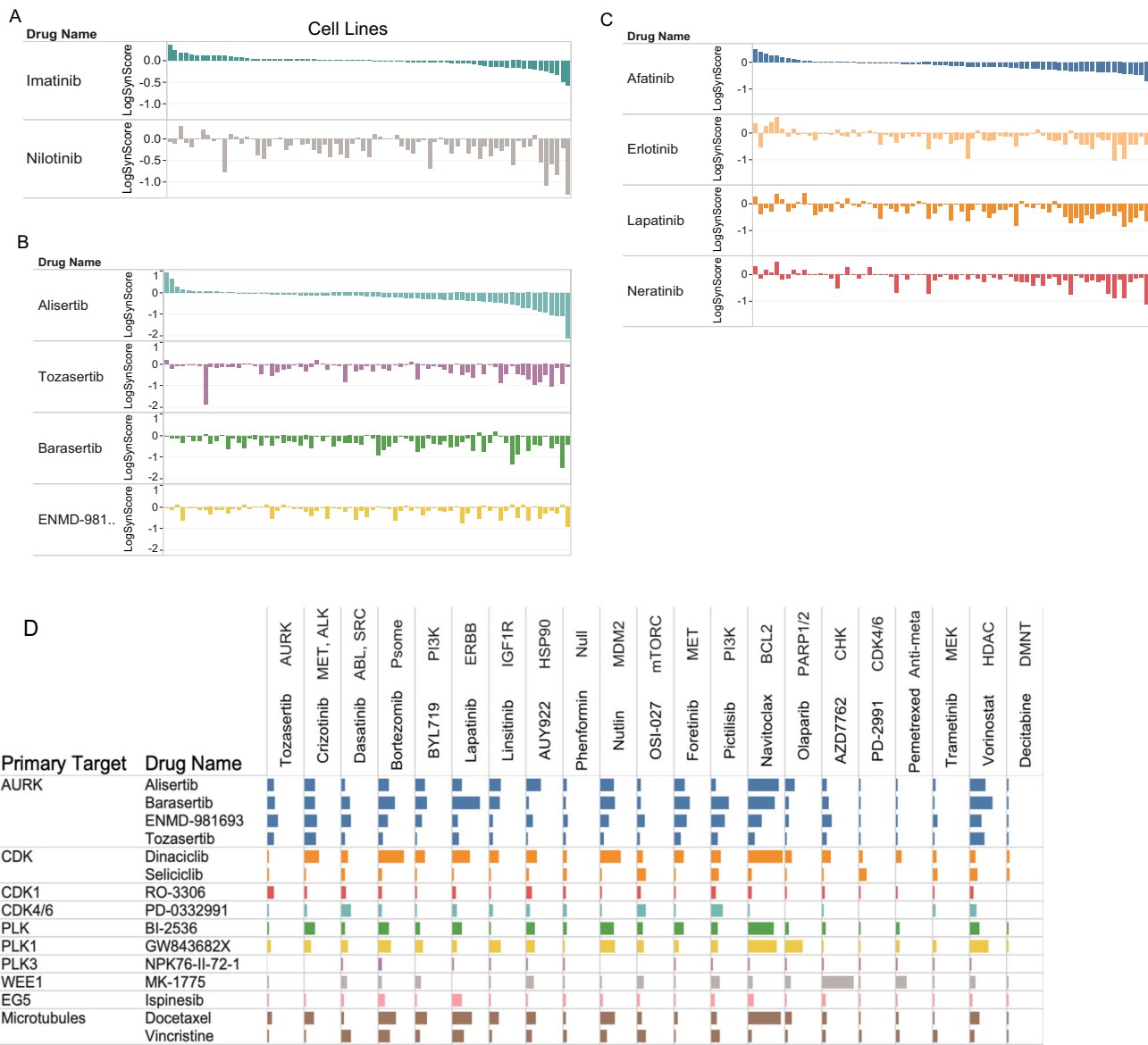

**Fig. 4 | Differential synergistic outcome for mechanistically related drugs. A–C** The pattern of synergies for a given drug is presented as a ranked ordered plot of synergy scores for the first listed drug (top) compared to the synergy score of related drugs with cell lines in the same order as for the first drug. The log of the synergy score is plotted. **A** Anchor drug is trametinib (MEK1/2) **B** Anchor drug is vorinostat (HDAC); **C** Anchor drug is OSI-027 (MTORC1/2). **D** Overview of differential pattern for related library drugs across anchors. Selected library drugs with similar targets were chosen (rows) and the number of synergies in combination with the indicated anchors (columns) are plotted as bars (the bar size is proportional to percent of the synergy scores that are within the top 5% of all synergy scores).

(termed the validation screen). We aimed at confirming some of the top synergies observed in the original screen (Fig. 3), as well as confirming the impact of polypharmacology on the synergistic potential (Fig. 4). We performed these studies on 15 non-small-cell lung cancer (NSCLC) cell lines chosen for genomic diversity. We tested 27 combinations that were present in the original screen. We also tested two combinations that were not present in the original screen to bolster our results using different compounds than in the original screen: Farnesyl transferase inhibition synergizing with MTORC inhibition and CHEK1/2 inhibition synergizing with WEE1 inhibition. Notably, to further probe the robustness of the original screen findings, the validation screen experiments were carried out in a different laboratory, using independently sourced stocks of cells and a different viability assay (CellTiterGlo) than the one used in the original screen experiments (imaging of nuclei). Additionally, the validation studies were carried out using full dose matrices with two ranges of concentrations tested

for each drug, resulting in four independent tests in 10 × 10 dosing format (Methods; Supplementary Data S8).

We first analyzed how the two screen datasets compared in terms of best combinations across the tested cell lines. The validation screened confirmed the observations of the original screen (Fig. 5, Supplementary Data S8; see Data availability section). Specifically, AZD7762 (CHK1/2) strongly synergized with the DNA damaging agent gemcitabine but not with another DNA damaging agent bleomycin, as seen in the original screen. AZD7762 also was confirmed to synergize in most cell lines tested with the WEE1 inhibitor adavosertib (MK-1775), as observed in the original screen. Furthermore, a second WEE1 inhibitor (PD-166285) also demonstrated strong and frequent synergies with AZD7762 and the patterns of synergies across cell lines were well correlated between the two combinations (Fig. 5). The ROCK inhibitor GSK-269962A was also confirmed to synergize with AZD7762 in a subset of cell lines. Confirming that inhibition of METAP2 synergizes

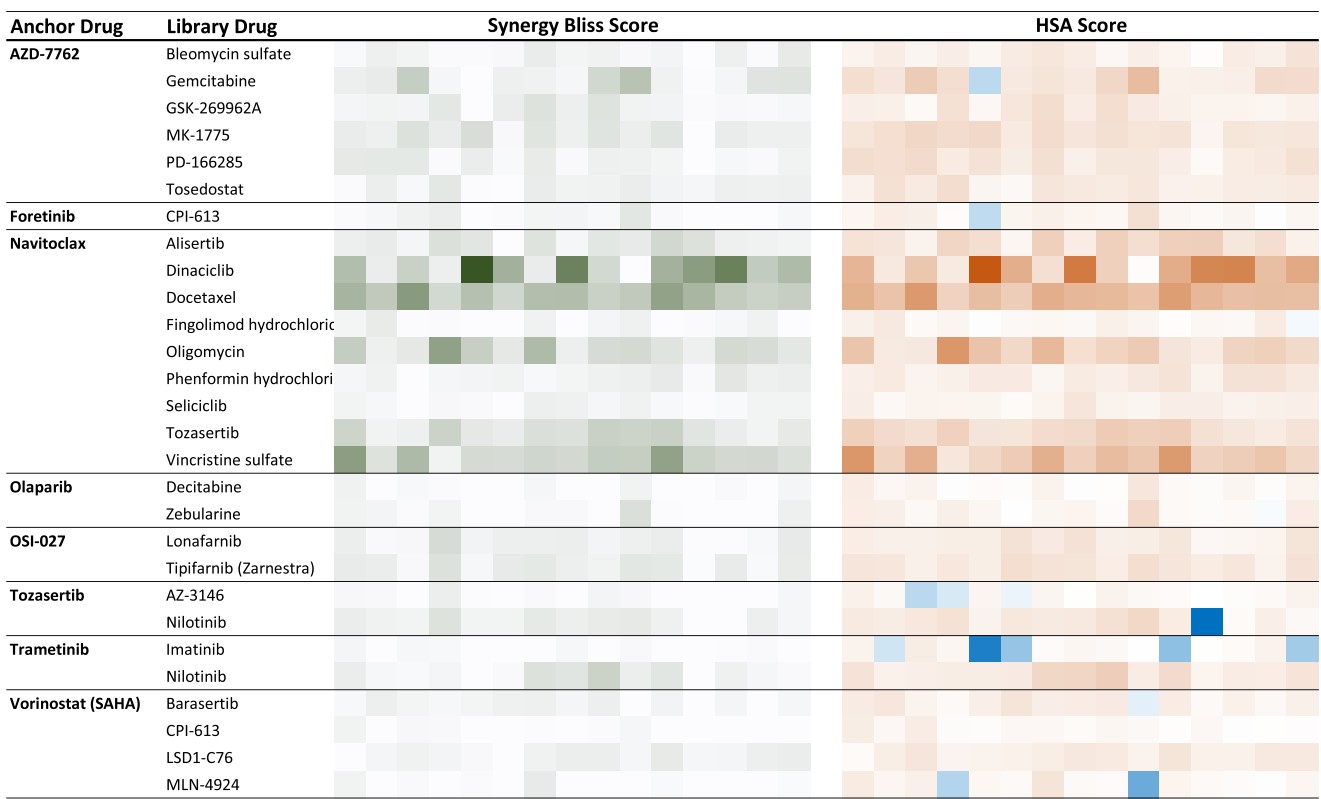

**Fig. 5 | Summary of the validation screen results.** Eight anchor drugs from the original screen were combined with a specific set of library drugs as indicated. The synergy score (Bliss score, green gradient, left section as labeled on the top) and HSA score (Orange gradient, right section as labelled on the top) is shown for each combination tested. Each column of colored squares corresponds to one of the 15 cell lines tested.

with inhibition of CHK1/2 as observed with A832234 in the original screen, tosedostat another METAP2 inhibitor was seen to synergize with AZD7762 in the validation screen. Vorinostat (HDAC inhibitor) was confirmed to synergize with barasertib (AURKB) and to a lesser extent with CPI-613 (PDH) MLN-4924 (NAE) and LSD1-C76 (LSD1), as seen in the original screen. In the original screen, the BCL2 family inhibitor navitoclax synergized with numerous drugs with diverse modes of action but often impacting cell cycle processes: Dinaciclib (CDK), alisertib (AURK), docetaxel (tubulin), oligomycin (ATP synthase), phenformin (ETC). All these results were confirmed in the validation screen. In addition, in the original screen seliciclib (CDK) was not as frequently synergistic with navitoclax than dinaciclib (CDK) and this was confirmed as well. Vincristine (tubulin) was more frequently synergistic with navitoclax in the validation screen than in the original screen where synergies were less frequent than with docetaxel (tubulin). Despite these differences, the synergy patterns across cell lines for docetaxel and vincristine were well correlated in the validation screen. Similarly, tozasertib was broadly synergistic with navitoclax in the validation screen even more so than in the original screen. We confirmed that olaparib (PARP) could synergize with zebularine and decitabine, this was less impressive in the validation screen than in the original screen, perhaps due to a shorter duration of treatment in the validation screen (2 days) than in the original screen (5 days), allowing for better capture of DNA damage effects in the original screen. The differential outcome of combining trametinib (MEK1/2) with the ABL inhibitors imatinib or nilotinib was confirmed as well, with synergies more frequent with nilotinib, likely underlined by nilotinib activity against RAF kinases absent in imatinib. A notable exception to the overall good concordance between the two screens, was the lack of synergistic activity between navitoclax and fingolimod in the validation screen. While perhaps this is due to the different readouts between the screens, the precise reason for this discrepancy remains

unclear at this point. More generally, ranking combinations based on the percentage of cell lines in which they were highly synergistic (Methods), demonstrate good correlation between the two screens outcome (Spearman's $\rho = 0.61$, $P = 0.00069$; Supplementary Fig. 9). We then analyzed how the two screens compared when considering which cell lines were most affected. We thus measured the correlation between the viability scores across all 15 NSCLC cell lines shared between the two screens for each of the 29 drug combinations. Compared to the high concordance between the two screens in terms of best drug combinations (above), here we found a more modest coherence across screens. We do observe a positive correlation between the two screens although the concordance is modest (12 and 15 drug combinations had a Spearman's $\rho$ and Pearson's correlation >0.25, respectively, Supplementary Data S10A, see top correlated examples in Supplementary Fig. 10). Notably, the concordance levels are quite similar when considering single agent treatment outcome. Because of specific growth conditions and potential non-genetic drift across batches of cell lines, we considered the possibility that some cell lines might present with less consistent results across the two screens. If we thus remove those cell lines which are overall less concordant between the two screens (Methods), the correlation indeed improves, and overall, 17 and 18 drug combinations had a Spearman's $\rho$ and Pearson's correlation >0.25, respectively; Supplementary Data S10B). As the two screens use different assays to measure the impact on cell viability, we would not necessarily expect very high concordance in this analysis. Nevertheless, these results point-out that while the very high-throughput screen performed on many cell lines robustly identify which drug pairs are most likely to synergize or otherwise more effectively affect NSCLC cells viability than the single agents, the pattern of drug response across cell lines is less stable.

Overall, the validation screen confirmed the robustness of the finding in the original screen both in terms of top drug pairs identified

as well as polypharmacology and differential synergy patterns observed between drugs sharing targets with common mechanisms of action.

Furthermore, to study how combinations might be broadly toxic across cellular lineages rather than selectively effective against NSCLC cells, we also carried out drug combination experiments on two non-cancerous immortalized cell lines (HUVEC and MDCK; Supplementary Data S8). We, therefore, tested the 29 drug combinations that we have studied above in 15 NSCLC cancer cell lines. Out of those, we identified nine drug combinations that are more effective (less viability) in at least 80% ($n \geq 12$) of the cancer cell lines than they are in the two immortalized cell lines (Methods; Supplementary Data S11A; eight out of them were found to be statistically significant (we analyzed the replicate data to carry out statistical significance tests (there were four replicates per drug combination in each cell line), using a one-sided Wilcoxon rank-sum test at FDR < 0.1). Repeating the analysis using synergy scores (instead of viability) showed that there are 11 drug combinations that are more synergistic in cancer compared to the immortalized cell lines (Methods; Supplementary Data S11B; 10 of them were found to be statistically significant, one-sided Wilcoxon rank-sum test at FDR < 0.1). Notably, three drug combinations were identified in both the viability and synergy-based screens, including Adavosertib and AZD7762, Nilotinib and Trametinib, and Imatinib and Trametinib.

We also mined drug combination data previously published in the DrugCombDB database[61], and found six NSCLC cell lines and 447 drug combinations overlapping with our dataset. We ranked all the drug combinations based on the number of synergistic cell lines in our cohort and in the DrugCombDB database and found that there is a marked correlation between the two ranked lists (Spearman's $\rho = 0.45$, $P = 6.4e-7$). In addition, an analogous analysis using the HSA measure showed good consistency as well (Spearman's $\rho = 0.54$, $P = 7.3e-10$).

We next compared the results of our drug combination screens to those obtained via drug combination experiments in NSCLC patient-derived tumor xenograft (PDX) models by ref. 62, which have tested five drug combinations across 36 NSCLC mouse PDX models. Although none of these drug combinations were exactly the same as those tested in our in vitro original drug combination screens, we were able to identify three drug combinations in Gao et al. that had similar drug targets to the drug combinations that we studied. These combinations show a similar percentage of high synergies or responders between the in vitro and in vivo data for PI3K and MEK inhibitors, PI3K and PIM inhibitors, IGF1R and MEK inhibitors (details in Supplementary Note 6). However, we do note that this analysis is quite indirect since we are only mapping drugs between the two datasets based on similar targets (as different drugs are known to have off-target effects).

## Emerging properties of combinatorial outcome

As discussed for specific examples of polypharmacology above, we find that combinations targeting just two targets (one single established target for each drug) are much less likely to be synergistic than combinations involving more than two targets ($P = 3.58 \times 10^{-34}$ one-sided Wilcoxon test). There is also a mild but highly significant correlation between the total number of targets involved in a combination and percentile of cell lines in which it is synergistic (Spearman's $\rho = 0.21$, $P = 6.25 \times 10^{-42}$), (Fig. 6A). Thus, drug specific effects are clearly seen and polypharmacology appears, as expected, to yield distinct context specific synergy outcomes, both at the drug and the cell-line levels. While this is likely to be an important hurdle for the rational development of combinatorial strategy, both in terms of efficacy and in terms of potential toxicities, it might also allow for the discovery of specific unexpected benefits (synergism) due to secondary target(s) inhibition.

A bird's eye view of the results of our screen (Supplementary Fig. 2) reveals that synergism typically occurs in a small number of cell lines and is thus strongly context dependent. Demonstrating that this sparsity of synergies is unlikely due to inappropriate dosing strategy, together with (Supplementary Figs. 1, 6B) shows that excessive dosing is not a likely broad cause of lack of HSA detection. Furthermore, the same sparsity property is seen with synergy scores. Because the synergy score is computed using a ratio of observed versus predicted outcomes, low viability outcome with single agents should not preclude detection of synergies. Consistent with this, there is only a very weak correlation between the viability outcome of the combination and the synergy score (Spearman rho = 0.038). To further study the sparsity of synergies across cell lines, the percentile of synergistic events for each anchor drug was computed defining strong synergy as the top 5% of all synergy scores observed in the screen. The HSP90 inhibitor luminespid has the most synergies (~7% of tests), followed closely by navitoclax. Pemetrexed and decitabine presented with the lowest number of synergies (below 2% of tests) (Fig. 6C). The drug-drug network corresponding to the top 5% synergy scores can be found in (Supplementary Fig. 8). The coverage of cell lines (proportion of cell lines presenting with synergy) was computed for different thresholds of synergy, and remain always sparse (Fig. 6D, E, Methods). A very similar pattern of distribution was observed when using a strong HSA score defined in an analogous manner (Fig. 6F, Methods).

An important corollary of the high level of sparsity of synergy events observed is that multiple combinations would likely be needed to provide potentially effective treatments to a cohort of many different patients. To address this, we computed the number of library drugs that need to be combined with each anchor drug to obtain strong synergy in at least 80% of the cell lines. If these results would carry to the clinic, this would inform how many drugs might be considered to combine with an established agent in order to improve outcome for most patients. Notably, this analysis revealed four anchor drugs for which a coverage of 80% could not be achieved regardless of the number of combination partners used (see Supplementary Note 2 for details). For the rest, the estimated number of drugs needed to obtain such coverage varied from 3 (bortezomib) to 18 (foretinib) (Fig. 6G). Supplementary Data S3a contains results for different coverage thresholds. We note that a 100% coverage was obtained for only one anchor drug (Navitoclax, 14 drugs needed in combination).

We sought to understand whether adding a second drug tended to make sensitive cell lines further sensitive or rather make resistant cell lines sensitive (or both). Analysis of the effect of combinations (viability) in relation to the sensitivity observed with single agents revealed that the combination outcome is almost always contained within the range of sensitivity ever observed with single agents (Methods, Fig. 7A). We term the rare drug pairs that diverge from this general pattern and actually yield an effect superior to what is seen with either agent alone in any cell-line "super-sensitizers" (Fig. 7B, Supplementary Data S3b, c). We see that a few drug combinations like OSI-027 (MTOR) and A770041 (LCK), vorinostat (HDAC) and tozasertib (AURK) show a supersensitive effect in more than 30% of the cell lines (Supplementary Data S3c). Individual anchor drugs, like dasatinib (BCR-ABL, SRC), olaparib (PARP), phenformin (Metabo), lapatinib (EGFR, ERBB2), vorinostat (HDAC) decitabine (DNMT1), show super sensitive effects in numerous different combinations. Across all library drugs tested, only a few are involved in supersensitive effects with any anchor drugs and a few of them are involved in supersensitive effects with more than one anchor drug (Fig. 7B). Super-sensitizers are highly enriched in synergistic pairs ($P = 1.14 \times 10^{-25}$, one-sided Wilcoxon rank-sum test) (Fig. 7C).

Leveraging synthetic lethality has been a major focus of target discovery and therapeutic strategy development in oncology for some time[63,64]. In the context of drug combinations, we define synthetic lethal (SL) interactions when neither of single agents are markedly effective, but the combination is (hence, these combinations correspond to a subset of more extreme synergistic interactions). To

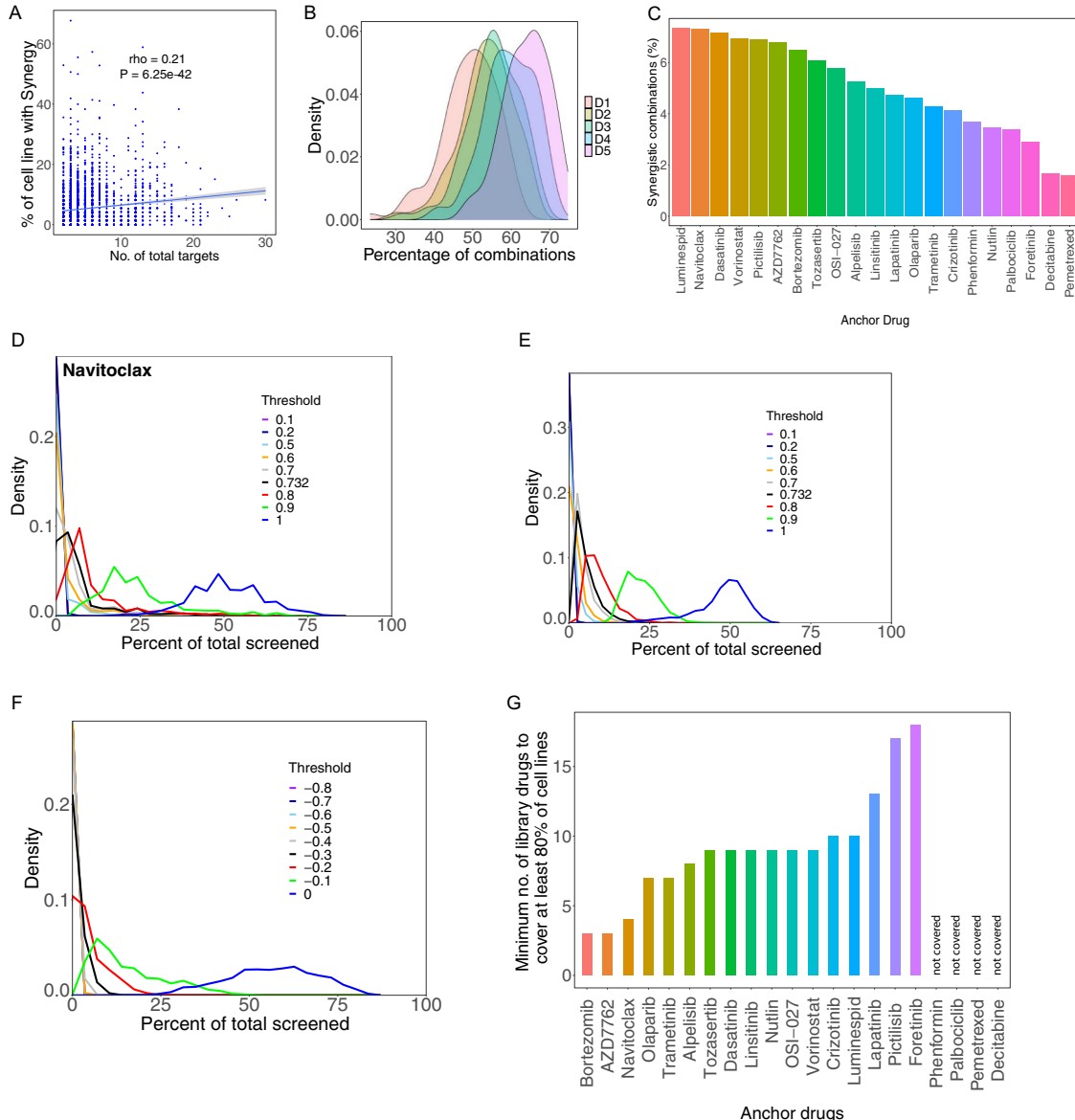

**Fig. 6 | Sparsity of synergistic and HSA events across models. A** The number of synergistic events for a given combination depends on the number of targets of the combination. The percentage of cell lines presenting with synergy for a given combination is plotted against the number of targets addressed by the two drugs together (number of total targets). Spearman's correlation (rho) and *p* values are shown. The gray band represents 95% confidence interval for predictions using linear model. **B** The proportion of cell lines harboring HSA increases with increased concentration of library drug. The density distribution of HSA events for each of the 5 doses of library drugs is plotted against the percent of cell lines presenting with HSA. **C** Percent of synergistic combination for each anchor drugs as determined by the percent of cell lines presenting with a synergy score falling within the top 5% of all synergy scores. **D** Density plots showing the fraction of synergistic events for anchor drug navitoclax for various thresholds of synergy (see Methods for details). A low number of synergistic events is consistently observed using several synergy score thresholds. **E** Same as (**D**) but across all anchors. **F** Same as (**E**) but considered HSA events instead of synergistic events (Methods). **G** Approximation of minimum number of library drugs needed to observe at least one synergistic event in at least 80% of the cell line collection.

quantify these SL effects in our screen, inactive single agents (and doses) are designated as those yielding a viability greater than 75% of control treatment (DMSO), but the combination is strongly effective (defined as below 40% viability). The number of such instances is very low, comprising just 1.32% of all possible combinations (at library dose D4, 2nd max dose; Fig. 7D). This result is reminiscent of the outcome of leveraging single agent sensitivity data in cell lines and combining single agent effective in a given set of cell lines to obtain synergy[65]. Interestingly, this result is comparable to fraction of synthetic lethal pairs seen in yeast and human cell line screens[66–68]. While rare, these drug combinations are potentially of exceptional interest from a translational point of view (Supplementary Data S3d). Some combinations like AZD7762 (CHK1/2) and adavosertib (WEE1) show

substantial viability effects even though their individual drug effects are weak in many of the cell lines (Supplementary Data S3d). This is seen for a small number of anchors and library drugs, with overall very few library drugs involved in such an effect with more than one anchor drug (CPI-613, KU-60019, carfilzomib or leptomycin B for example) (Fig. 7D). We further note that while true synthetic lethality (as strictly defined above) underlies a small minority of the synergistic combinations, evidently, synergism is essentially equivalent to synthetic sickness.

To study the impact of the cell-line cancer driver genotype on drug combination outcome, the percentage of highly synergistic combinations (top 5% synergy rank) across different genotypes was analyzed. Overall, across all anchors and drugs, there was no statistical

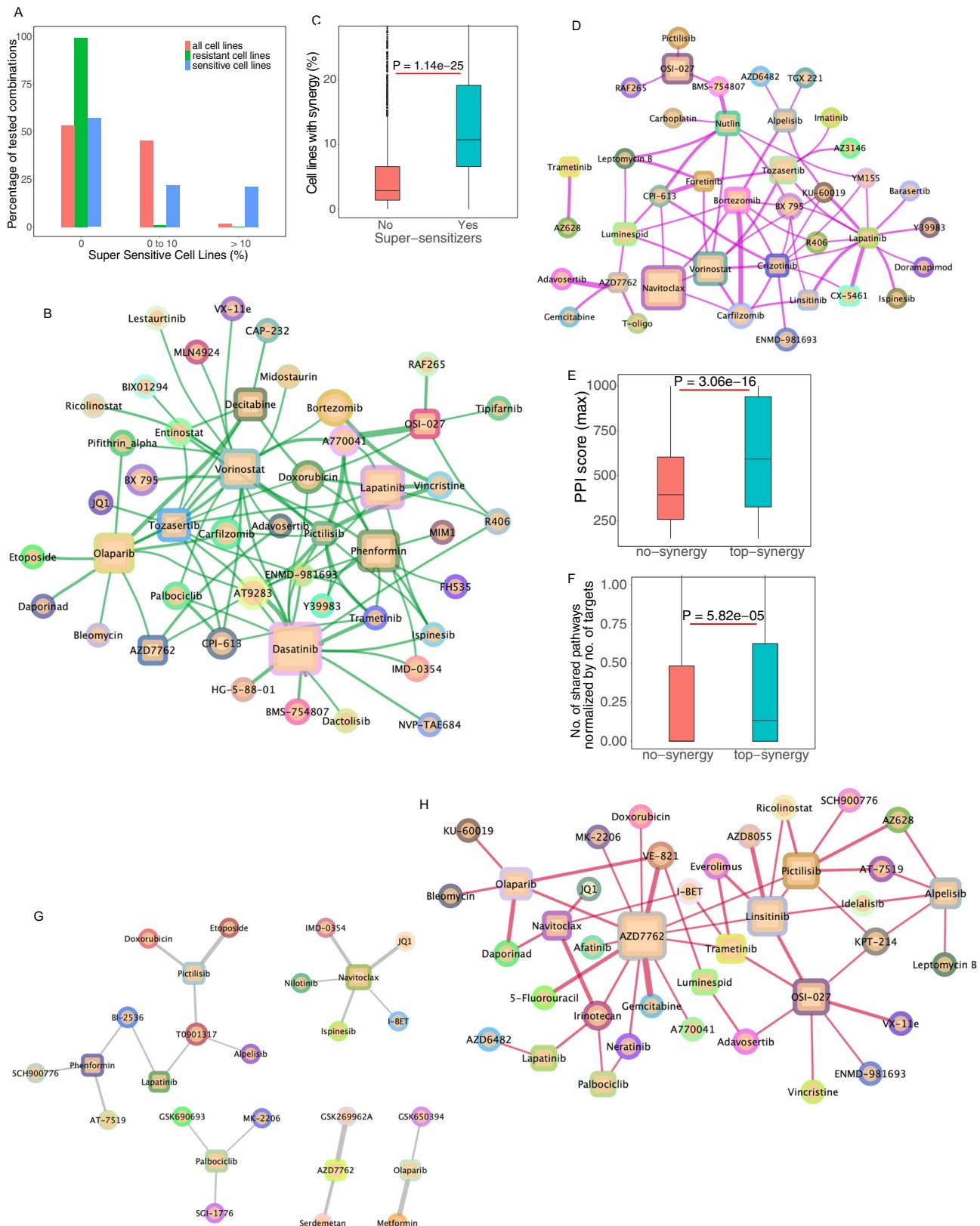

imbalance for synergism for any of the major cancer driver genotypes, including *KRAS*, *PIK3CA*, *EGFR*, *STK11*. Interestingly, among KRAS mutant cell lines, *STK11* (encoding LKB1) mutant cell lines were seen to harbor more synergies than *STK11* WT ones. Although TP53 encodes a major tumor suppressor and sensor of cellular stress, *TP53* mutations were not associated with synergism (or lack thereof). This is

reminiscent of the results obtained with single agent treatment of very large cell line collections, where *TP53* mutational status is not a strong predictor of drug response[14,15]. The analysis of the genetics of synergism for each anchor individually revealed that three anchors, crizotinib (MET / RTK), phenformin (ET Complex V) and AZD7762 (CHK1/2) were statistically more synergistic (number of top 5% synergies) in the

**Fig. 7 | Relationship between sensitivity to single agents and combinatorial outcome. A** Combinations rarely affect viability beyond effects observable with single agents. The effect of combination treatment on each cell line was compared to the overall sensitivity to single agents observed across all cell lines. The percentile of cell lines in which the combination effect is superior to the effect of single agents in any cell lines (super sensitive cell lines) is shown in three categories (0, no observable supersensitive lines, 0–10% and over 10%). The cell lines are further broken down based on their response to single agents (color). **B** Network view of drug combinations resulting in super-sensitization. Anchors are displayed as squares and library drugs as circles. Library drugs that are also used as anchor drugs are represented by squares. **C** Supersensitive events are enriched in synergies. The percentile of synergistic events (cell lines) are compared between combinations that yield super sensitization versus ($n = 92$) those that do not ($n = 4990$). One-sided Wilcoxon rank-sum test $p$ value is shown. **D** Synthetic lethal drug pairs: Network of drug combinations for drugs that yield substantial viability effect while single agents are deemed inactive. **E** Drugs with targets that are in close interaction with each other based on previously determined protein–protein interaction network are more likely to yield synergies than those targeting un-connected proteins ($n = 157$ 'top-synergy', $n = 58$ "no-synergy"). One-sided Wilcoxon rank-sum test $p$ value is shown. **F** Drugs that target members of a given biological pathway are more likely to yield synergy than those that target members of different pathways ($n = 635$ 'top-synergy', $n = 784$ "no-synergy"). One-sided Wilcoxon rank-sum test $p$ value is shown. The box plots for figures (**C**), (**E**), (**F**) have median center, 25 and 75 percentiles (Q1 and Q3) as bounds of the box, the minima and maxima being Q1 − 1.5 x IQR and Q3 + 1.5 x IQR (with IQR being interquartile range). **G** Synergistic drug pairs for which targets are encoded by genes engaged in a synthetic lethal interaction based on TCGA data analysis. **H** Synergistic drug pairs with experimental evidence for a synthetic lethal relationship.

*KRAS* mutant than in *KRAS* WT (one-sided Wilcoxon test, FDR < 0.2). By contrast, six anchors promoted less synergies in *STK11* mutant cell lines (*KRAS* WT and mutant) than *STK11* WT ones: Tozasertib (VX-680, AURK), linsitinib (OSI-906, IGF1R), luminespid (AUY922, HSP90), phenformin (ETC), nutlin (MDM2), foretinib (XL-880, MET/ RTK). The combination of Bortezomib (proteasome) and AZD8055 (MTOR) was found to be more synergistic (using second-best synergy measure) in *KRAS* mutant than wildtype cell lines (one-sided Wilcoxon test, FDR < 0.2). Additional results are presented in Supplementary Data S4. When considering combination effectiveness rather than synergism, a number of combinations were seen to be more effective in a given mutant genotype (library dose D4, the second max dose was used for this analysis) (Supplementary Data S4): 195 for *EGFR*, 1128 for *KRAS* and 158 for *PIK3CA* mutated drivers (one-sided Wilcoxon test, FDR < 0.2). As expected, most of the combinations showing higher effectiveness in *EGFR* mutant vs WT cell lines contained the EGFR inhibitor anchor lapatinib (144 out of 195). Similarly, for *PIK3CA* mutants, the PI3Kapha inhibitor BYL719 is present in all (158 out of 158) combinations showing higher effectiveness in *PIK3CA* compared to the WT. Interestingly, for *KRAS*, only 35 combinations involving trametinib showed an imbalance in effectiveness. Olaparib (AZD2281, PARP, 231 out of 1128), linsitinib (OSI-906, IGF1R, 150 out of 1128), navitoclax (ABT263, BCL2, 118 out of 1128) were anchors showing a high level of differential effectiveness in *KRAS* mutant versus WT models. Mutations in *TP53* were associated with lower effectiveness for 199 combinations of which 184 involved nutlin (one-sided Wilcoxon test, FDR < 0.2). This is expected since nutlin is predicably ineffective in *TP53* mutant cell lines[14]. Contrary to expectations, we find that *KRAS* models did not respond to combinations treatment significantly differently from the *KRAS* WT cell-lines. This observation is further confirmed by a principal component analysis of the post-treatment viability values showing that *KRAS* WT and *KRAS* mutant models do not segregate away from each other (Supplementary Fig. 11). We also looked for differentially synergistic and effective drug combinations between mutant and wild-type cell lines involving over 400 cancer driver genes (provided in Supplementary Data S1f). Although we could not find combinations that are differentially synergistic between mutant and wild-type cell lines, we found several combinations that were less effective in a given mutant in comparison to the wild-type genotype (FDR < 0.2): those include 684 and 217 drug combinations for *TERT* and *MACF1* respectively (Supplementary Data S4). For the mutant vs wildtype comparison for the *TERT* gene, the most differential anchors were AZD7762 (CHK) and pictilisib (PI3K); they appear in 196 and 163 out of 684 combinations, respectively. For the *MACF1* gene, the most differential anchor was AZD7762 (appears in 196 out of 217 combinations).

Overall, the brunt of the synergistic events is hence not accounted for by the mutational state of recurrent cancer driver genes. This is in keeping with previous studies of drug combinations reporting a rather idiosyncratic pattern of synergy across models tested when classified on genotype alone[13,35,36,69,70]. Additionally, there was little difference in

synergism or efficacy across subtypes of NSCLC (squamous, adenocarcinoma). Phenformin was the only anchor showing a subtype imbalance, with adenocarcinoma models harboring more synergies than squamous cell carcinoma models ($P = 0.00845$, FDR < 0.2, one-sided Wilcoxon test).

To gain a more general view of the characteristics of the targets involved in synergistic drug pairs, the protein–protein functional interaction (PPI) database STRING[71] was queried. This revealed that the targets of highly synergistic drug pairs are closer in the PPI network than the targets of non-synergistic drugs ($P = 3.06 \times 10^{-16}$, one-sided Wilcoxon rank-sum test) (Fig. 7E). The same outcome was obtained using either all evidence or only experimentally validated PPIs in STRING and using median or maximum PPI score across targets. In an analogous manner, analysis of the KEGG pathway database[72,73], showed that combinations targeting proteins within the same rather than different pathways are more likely to be synergistic (corrected for total number of targeted pathways to compensate for polypharmacology, see Methods, $P = 5.82 \times 10^{-05}$, one-sided Wilcoxon rank-sum test) (Fig. 7F), in accordance with previous findings[74–77].

To further evaluate the potential clinical relevance and benefit of the synergistic drug combinations identified here, a Cox regression analysis was performed using patient tumor data in TCGA. After controlling for single gene effects, age, sex, race, and cancer type among the 981 TCGA NSCLC samples, the concomitant down regulation of the genes composing 43 drug combinations (6.04%), were associated with an improved patient survival ($P < 0.05$, Methods, Supplementary Data S5a). The BCL2-inhibitor navitoclax appears in 14 of these combinations. Repeating this analysis using copy-number data, showed that for 33 drug combinations low copy-number of the corresponding target pairs lead to improved predicted survival (Supplementary Data S5b). Here, the BCL2-inhibitor Navitoclax appeared in 17 combinations. We note that this survival analysis only provides limited correlative support and obviously, should not be taken as causal evidence. We did not find an enrichment of highly ranked synergies for targets whose downregulation is associated with increased survival benefit in NSCLC patients (Fisher exact test, odds-ratio = 0.99, $P = 0.56$).

To uncover what fraction of the synergistic combinations identified in the screen might arise from SL interactions between their targets (this conceptually differs from the previous analysis presented above, where we quantified the direct SL-like interactions between the drugs themselves based on their phenotypic reduction of cell viability), we employed the ISLE pipeline[75] to analyze the lung cancer patient cohort of TCGA and identify the SL partners of each of the targets of the drugs screened in our analysis. Among the 1166 SL pairs found (Supplementary Data S6a, Methods), 83 drug combinations were seen to be linked via their targets to any of these SL pairs. (Supplementary Data S6b). Notably, among those, 21 drug combinations were synergistic (top 25% of synergy scores, Supplementary Data S6c, Fig. 7G). Some synergistic combinations like AZD7762 (CHK1/2) and GSK269962A (ROCK), navitoclax (BCL2 family) and JQ1 (BRD) target

genes that are predicted to be synthetic lethal based on the TCGA data analysis. Anchors like navitoclax, phenformin, pictilisib (PI3K), palbociclib (CDK4/6) are involved in a few high synergy combinations that target these synthetic lethal interactions, T0901317 is the only library drug that is seen to have this relationship with more than one anchor (Fig. 7G). We did not find an enrichment of top ranked synergistic combinations among the combinations that have synthetic lethal targets (Fisher exact test, odds-ratio = 1.2, $P = 0.27$). However, we believe that these additional down-stream analyses are still of interest, as they identify the subset that is also associated with survival effects, as another potential indicator (albeit a very rough one) of their putative translational value. Beyond that, we note that the lack of enrichment is not very surprising, given the well-known discord between in vitro and in vivo studies (e.g., refs. [78–80]) and that not all drugs that are beneficial will necessarily show a survival benefit (e.g., ref. [81]).

Further, comparing the screened combinations to a published dataset of SL pairs identified in cell lines[82] (Supplementary Data S6d), showed that for 155 targets, at least one SL pair exists (Supplementary Data S6e) and that 55 of them are synergistic (Fig. 7H, Supplementary Data S6f). Multiple synergistic combinations, including AZD7762 (CHK1/2) and Gemcitabine (DNA synthesis antimetabolite), AZD7762 and VE-821 (ATR), target genes that are synthetic lethal based on experimental evidence. Some anchors like AZD7762, Pictilisib and OSI-027 are involved in several high synergy combinations that target such SL interactions and overall, a small number of library drugs are involved in synthetic lethal relationship with multiple anchors (Fig. 7H). Five drug combinations were common to both tumor derived and experimentally derived SL pairs, of which 2 were synergistic: Navitoclax + I-BET and navitoclax + JQ1, that are mapping on the same targets (I-BET and JQ1 are both BRD targeting drugs). Thus, overall, SL analysis in patient data can explain a relatively small subset of synergistic combinations but might be useful to prioritize combinations found in the screens, indicating that they may be clinically relevant.

## Discussion

In this manuscript, we describe the outcome of a very large combinatorial drug screen surveying over 5000 two drug combinations across 81 NSCLC highly characterized cell lines, some of which are further tested in an additional validation screen. By mining the literature on published drug combinations and using prior knowledge of cellular circuitry, we demonstrate the validity of both the data and the analytical strategy. Overall, we capture a large number of known or mechanistically transparent synergistic events that are consistent with prior knowledge as well as a number of less characterized ones. A subset of those have support from synthetic lethal analysis of NSCLC patient tumors data, suggesting potential translational relevance.

To inform the robustness of the screen outcome we performed a small validation screen using a different viability assay than our massive primary screen. Furthermore, we performed the validation screen in a different facility than the primary screen. Consistent with our findings that each anchor drugs synergizes strongly with the expected library drugs across 81 cells lines, we observe that most frequently synergistic combinations are re-captured in the validation screen. Altogether, these results support robustness of our primary dataset. However, we also observe that cell lines presenting with synergy in the primary screen do not always match the cell lines in the validation screen. This is a limitation of our results and is in keeping with analyses performed on robustness of screens across independent studies[83] and likely further exacerbated by the use of different viability assays.

To prioritize drug combinations that are less toxic, we tested the same 29 drug combinations used in the validation screen on a couple of immortalized non-cancerous cell lines and identified a subset of them that are selectively effective or synergistic in cancer cell lines. However, further in vivo experiments are required to evaluate their toxicity and further prioritize combinations for translational

investigations, given the obvious limitations of using cell lines for toxicity studies.

One of the most striking outcomes of our analyses is that synergistic combinations are mostly sparse and thus highly context specific. This is in line with the findings recently reported studying a large drug combination screen of breast, colorectal, and pancreatic cancer cell lines[84]. Interestingly, some of the top synergistic combinations identified in the present work were also present in other cancer types studied in the Jaaks et al. study[84]. Of note, synergy between CHK1/2 inhibition and WEE1 inhibition is common across models from diverse tumor types. More generally, we find that combining drugs that are not active as a single agent almost never yields synergy. In addition, combining two drugs tends to render single agent resistant cell lines responsive rather than further sensitize already sensitive cell lines. Furthermore, sensitive cell lines rarely become super-sensitive, as combination effects mostly fall within the minimum viability levels observed for their individual components across all cell lines. While this could potentially correspond to limited efficacy of combination of agents that are not efficacious on their own, exceptionally sensitizing combinations can be found. In addition, synergism might more broadly provide benefit by allowing context specific activity of lower drug doses than used with single agents. The mutational status of major cancer genes is not highly predictive of synergy, as observed in other studies. Finally, synergy is more likely to emerge from targeting a single pathway or two interacting pathways, than by targeting two completely distinct pathways or functional modules of the cell. This finding is aligned with previous reports based on studying genetic perturbations in lower organisms[74]. One potential model explaining these findings is that when two combined drugs target sufficiently independent cellular functions then the highly evolved and robust homoeostatic control of the cellular system prevails.

There is still considerable debate over what is synergy. Several competing models, that nevertheless often yield congruent conclusions[26] are used to qualify and quantify synergy. Here, a synergy scoring based on statistical independence akin to the broadly used Bliss model was used. We and others have previously demonstrated that this model is indeed valid to study viability outcome upon combinatorial treatment[35,85]. Nevertheless, it is often pointed out that this type of modeling can in some instances assign synergy to cases of self-additivity. It is important to note that this counter intuitive outcome is limited to a small number of drugs. Examination of the relationship between self-synergy paradox and dose response shows that the drugs concerned have very steep dose response curves. Indeed, as seen here, vorinostat for example, has a much steeper dose response curve than most other drugs (see Supplementary Note 4 for details). Our results indicate that synergy is overall a rare event, thus most drug combinations are explained by the independent action of the two drugs combined (as explicit in the Bliss hypothesis). This is aligned with recent modeling of clinical combination effectiveness[86] and their historical empirical development in cohorts of molecularly heterogeneous patients[4].

Supersensitive combinations compose a different entity than synergistic combinations. The former denotes a specific class, where the combination achieves a response (in a subset of cell lines) that is higher than the highest response of any of the two drugs composing it observed across any of the cell-lines. Synergistic combinations, in difference, are combinations that achieve better response than the individual drugs composing them in many cell lines, but not necessarily better than the best individual cell-line response. Our interest in supersensitive combinations has been specifically motivated by recent work presenting the Independent Drug Action (IDA) principle[86], which has claimed that the vast majority of combinations fail to achieve a response that is higher than the highest response of any of the individual drugs composing them. From a translational standpoint, we would like to identify combinations that are not only synergistic (that

is, better than the mean additive effect of the individual drugs across cell-lines), but also supersensitive (that is, better than expected across all cell-lines). Though the two classes are overlapping to some extent, we believe that each captures a different and important notion from a translational standpoint.

In summary, this work presents and analyzes the results of a very large dataset of drug combinations across lung cancer. The resulting dataset, that we make fully accessible to the scientific community, substantially expands on previous publicly available resources for drug combinations mining and modeling. There are many more analyses that could be performed using the data herein. Beyond that, our hope is that these data will foster the development of additional analyses and novel computational approaches advancing the prediction of drug-drug combinations outcome and our understanding of the rules underlying beneficial drug interactions in cancer.

## Methods

### Drug screening and cell viability determination
Cell line sources are listed in (Supplementary Table S3). Drug screening was performed using automated liquid handling in a 1536-well plate format. The drug doses used were chosen based on previous single agent screening at the Center for Molecular Therapeutics of the Massachusetts General Hospital Center for Cancer Research. The screen of two drug A and B was performed in a 1 × 5 format with one dose of drug A (anchor drug) combined to five doses of drug B (library drug) and compared to the effects of the five doses of drug B alone. The five doses of drug B followed a √10 dilution series. Screening was performed in replicate (two separate 1536 well plates).

Effect of drug treatment was determined by enumerating cell nuclei 5 days after the addition of drugs (day 0 designate the seeding day and day 1 the drug treatment day; no change of culture medium or drug re-addition were performed). Cells were seeded at densities optimized for proliferation based on pre-screen experimental determination in 1536 well plate format.

Cells were seeded, placed overnight at 37 °C and drugs added the next day using a pin tool.

After 5 days in drug cells were fixed permeabilized and the cells' nuclei stained in a single step by adding a PBS Triton X100/Formaldehyde/Hoechst-33342 solution directly to the culture medium. Final concentrations: 0.05% TX-100/1% Formaldehyde/1 ug/ml Hoechst-33342. Plates were covered and placed at 4 °C until imaging. Imaging was performed on a ImageXpress Micro XL (Molecular Devices) using a 4 × objective. Cell nuclei enumeration was performed using the MetaXpress software and count accuracy was routinely checked visually during acquisition.

No washing of plates was performed at any point post seeding.

Viability was computed as the ratio of number of nuclei in the drug treated wells over those in the control (DMSO treated) wells. For the drug combination plates, the anchor drug was added to all wells. The relative viability (compared to anchor alone treatment) was then computed by dividing the number of nuclei in treated (drug combination wells) by the anchor drug alone wells. This viability was then compared to the viability computed from the single agent wells (DMSO as an anchor). This allows for direct comparison of drug effect without using the values of the DMSO only wells (in the DMSO anchored plate) to compute drug combination effect. While mathematically equivalent to the cross plate comparison, this approach allows to minimize data noise due to potential plate to plate cell seeding number variation. Quality control criteria included a CV of less than 25% of the control wells (either DMSO alone wells for the DMSO anchored plates or Anchor alone in the Combination plates) and a cellular proliferation of at least one doubling. We combined replicates by taking the median values of the viability scores for each cell line and drug combination for each library drug dose. Proliferation was computed by comparing Day1 untreated plates (seeded concomitantly

with the assay plates and fixed the day after seeding) to the DMSO only wells in the Day 6, DMSO anchor plates (Assay plates).

### Highest single agent (HSA) scores and synergy scores
We compute the HSA effect for each drug combination (on a cell line for a particular dose) by subtracting the combination viability minus the best (minimum) viability of the corresponding individual drugs. A negative value implies that the drug combination is more effective than the better of the two individual drug effects.

Synergy scores are computed using the Bliss model (Goldoni and Johansson, 2007; see Supplementary Note 1 for details). The lower the score, the more synergistic the drug-combination is. A drug combination is synergistic if its score is less than one. For each anchor-library-cell-line combination, we also compute the second-best synergy score among all the five library doses. We defined a drug combination to be highly synergistic in a cell line, if its synergy score (second-best, i.e., second-lowest) is less than a certain percentile (for example, five percentile or value of 0.732) of all combinations in all cell-lines. We show our results using various thresholds for detecting high-synergy combinations. We also compute the percentage of cell lines which are highly synergistic for each combination, and then rank all drug combinations based on this measure.

We computed empirical $p$ values for synergy scores for each drug combination at the five library drug doses and the second-best synergy score by converting the log-transformed synergy scores (to make the synergy ratios more normal distribution like) into z-scores and then computing empirical $p$ values assuming a normal distribution. Similarly, empirical $p$ values were also computed for HSA values for all cell lines and combinations (Supplementary Data S7).

To see if the replicates agree on the strength of synergy, we checked if the synergy scores obtained between the two replicates are correlated, by computing the Spearman's correlation between their synergy scores across all cell lines and doses. We find a very significant positive correlation between the two replicates for each one of the 21 × 242 drug combinations (Supplementary Data S9).

### Super-sensitizers
For each drug combination, we score the cell lines based on the max of the two drug effects (minimum viability). The ten lowest ranked cell lines are considered to be resistant cell lines for the combination. The ten top ranked cell lines are considered to be sensitive cell lines for the combination (excluding the most sensitive cell line). Now we check if the combination response (for library dose D4) in the resistant/sensitive cell lines is better than the best individual drug effect in the most sensitive cell line. If the combination effect is indeed better, we say that the combination makes the cell line super-sensitive. We considered all combinations which show super-sensitizer effect in at least 10% of the cell lines, and called them super-sensitizers.

### Drug-target mapping
We mapped the drugs to their targets using several resources: DrugBank[87], Selleckchem.com. The mapping is shown in Supplementary Data S1c, d.

### Protein−protein interaction (PPI) scores
We downloaded PPI network scores from the STRING database[71] (downloaded on Aug. 8, 2019). We computed the PPI interaction score between the drug targets of any two drug combinations. If drugs have multiple targets, we either compute the max or median PPI score between the respective drug target pairs. We did this analysis using both the entire PPI network and by considering only drug target pairs which are bound to each other (called 'binding' in STRING).

The PPI score for the drug targets of the top synergistic combinations (based on top 5% synergy score overall) was computed based on all PPI information types in STRING. The PPI scores for the synergies

with high scores in at least 10% of the cell lines tested (637 combinations) were compared to those for the non-synergistic pairs (lacking any strong synergy across cell lines, 743 combinations). For multi targeted drugs the maximum PPI score across targets was considered. This is for the analysis in Fig. 7E. The same results were obtained using only experimental binding evidence for PPI in STRING or using the median PPI score across targets rather than the maximum across targets.

### Patient survival analysis and synthetic lethal (SL) analysis

To test whether clinical survival benefit could potentially be derived from treatment with synergistic drug combinations identified, we mined data from 981 TCGA NSCLC patients (lung adenocarcinoma and lung squamous cell carcinoma patients). Only combinations whose targets could be mapped to TCGA gene set and with less than four targets per drug were considered because a search across a large number of targets might show a survival signal by chance. There were 712 such drug combinations among the top 25% most frequently synergistic combinations. For these combinations, low expression (below 33 percentile) of at least one of all potential target pairs yielded an improved survival benefit after controlling for single gene effect, age, sex, race, and cancer type among the 981 TCGA NSCLC patients (both lung adenocarcinoma and lung squamous cell carcinoma patients[88,89]. The assumption being that the down-regulation of the target pair(s) may simulate clinical administration of the combination.

We used a computational method called ISLE[75] which mine 981 TCGA NSCLC patients (lung adenocarcinoma and lung squamous cell carcinoma patients) to identify clinically relevant SL pairs. ISLE uses four different filters: (a) It firsts mines in vitro shRNA/CRISPR datasets spanning hundreds of cell lines to identify potential SL candidates; (b) It then looks for negative selection of co-inactivated gene pairs using gene expression and copy number analysis in TCGA cancer patients; (c) ISLE then selects gene pairs whose co-inactivation (low expression or copy number) is associated with improved survival; (d) It finally selects SL pairs where the genes have high phylogenetic similarity. More details of this method is explained in ref. 75. FDR threshold of 0.2 was used for this analysis. We mapped the drug combinations to their targets and mined for clinically relevant SL interactions between these target pairs. Only drug combinations where both the individual drugs have less than four targets are considered for this analysis. We identified experimentally derived SL gene pairs from various studies. The compilation of these various studies is provided in ref. 90. There are 27975 experimentally identified SL pairs (Supplementary Data S6d).

In both these analyses, a highly synergistic drug combination in a given cell line is by considering the top 5 percentile.

### Analysis for Fig. 5D−F

For a fixed threshold of synergy or HSA, for each library drug at some dose, we look at all the anchor and cell line combination and check the fraction of them below the fixed synergy/HSA threshold (percentage of high HSA or high synergy). This will be our fraction of high synergies/HSAs for that library drug at some dose. We plot a density of these values for the fixed synergy/HSA threshold. We repeat the above procedure for different thresholds of synergy and HSA. We can also do the above procedure for a particular anchor drug.

### Validation screen: quantitively high-throughput and matrix screening

All cell lines except for H2009 and A549 were purchased from American Type Culture Collection (ATCC, Manassas, VA). H2009 and A549 were obtained from internal NCATS stock. Cell culture conditions were as follows:H727, H1437, H1666, H1703, H1734, H1755, H1915, H2347 and H2405 cells were grown in RPMI1640 medium (ATCC 30-2001); A549 cells were grown in F12K medium (ATCC 30-2004), A427, Calu-6 and SK-Lu-1 cells were grown in EME medium (ATCC 30-2003); H1651 and

H2009 cells were grown in DMEM: F12 medium (ATCC 30-2006). All media was supplemented with penicillin/streptomycin and 10% fetal bovine serum (FBS). All the celllines were maintained at 37 °C in a humidified $CO_2$ incubator. For pairwise drug-combination assessments in matrix format, compounds were acoustically dispensed into a 1536-well white solid bottom tissue culture-treated plate (EWB041000A, Aurora Microplates, Whitefish, MT, USA) with an Echo 550 acoustic liquid handler (Labcyte, San Jose, CA, USA). A 9-point custom concentration range with 1:2 dilution between points was used for each drug-pair tested. Bortezomib (final concentration 20.3 μM) was used as a positive control for cell cytotoxicity. Cells were seeded into compound-containing plates at a density of 500 cells/well, in a final volume 5 μL of growth media by using a Multidrop Combi dispenser (Thermo Fisher). Plates were covered by a stainless-steel gasketed lid to prevent evaporation and incubated for 48 h in a humidified CO2 incubator. At the 48-h time point, 3 μL of Cell Titer Glo (Promega) was added to each well using a Multidrop Combi dispenser and plates were incubated at room temperature for 15 min with the stainless-steel lid in place. Luminescence readings were taken using a Viewlux reader (PerkinElmer) with a 2 s exposure time per plate. Viability of compound treated wells was normalized to DMSO and empty well controls present on each plate, and combination-response plotting was automatically performed for each individual drug+drug combination.

There are 29 drug combinations and 15 NSCLC cell lines in the validation screen, out of which there are 27 common drug combinations between the original and validation screen. The validation experiments were done in 10 doses × 10 doses in 4 different dilution series. high maximum × high maximum, low maximum × low maximum; high maximum × low maximum, low maximum × high maximum. From this HSA score and synergy score was computed for each. Because of this, the overall robustness of the outcome on a given combination is high and we take the best overall score for each cell line drug combination. An Excess_HSA score of less than −1000 was chosen as threshold to designate best combinations. The cell-line drug combination in the original screen was chosen to be highly synergistic if its second-best synergy score was in the top 20 percentile among the combinations considered for this analysis. Each of the common drug combinations (in both the original and validation screens, separately), were ranked based on the percentage of cell lines (15 cell lines) in which they were highly synergistic. Spearman's correlation between the ranking of the drug combinations in both the original and validation screens was then performed.

For the 29 drug combinations (out of which there are 27 common drug combinations between the original and validation screens, and the remaining two combinations are matched based on similar drug targets), we tested for the concordance of drug combination sensitivity between the original screen and validation screen by measuring the correlation (both Spearman's and Pearson's correlation) between the viability scores across all 15 NSCLC cell lines between the two screens. In the original screen, as in most of our analysis, we considered library drug dose D4. For the validation screen, we considered the closest anchor and library dose concentrations that correspond to the corresponding doses used in the original screens. A similar analysis was also done when quantifying and analyzing the individual drug response. We also repeated the concordance analysis by removing outlier cell lines and considering only cell lines having a Spearman's $\rho > 0.25$ between the original screen and validation screens (the pertaining results are reported in the main text).

For the toxicity analysis, we tested the 29 drug combinations on two non-cancerous cell lines in addition to the 15 cancer cell lines. We have four block IDs that map to each drug combinations. For each block ID, we took the median viability score by considering all (10 × 10) doses. Then for each combination, out of four block IDs, we took the best (minimum) viability for every cell line. This forms the viability score for each drug combination in each cell line. We then identified

drug combinations that are more effective (using viability scores) in at least 80% ($n \geq 12$) of the 15 NSCLC cell lines in comparison to their effectiveness in the non-cancerous cell lines. A similar analysis was done for synergy (ExcessHSA score). For computing statistical significance of either more effective or synergistic drug combinations in cancer cell lines in comparison to non-cancerous cell lines, we considered all the four replicates (block IDs) and computed one-sided Wilcoxon rank-sum test between viability or synergy scores between cancer and normal cell lines (FDR < 0.1).

### DrugCombDB analysis

Matching the drug names and cell line names in our cohort with the DrugCombDB database[61], we find 6 NSCLC cell lines and 447 drug combinations that match. Whenever there were any replicates in the DrugCombDB database we took the median synergy or HSA score across replicates. For each drug combination, we ranked the combinations by the number of cell lines which are synergistic for that combination (median synergy score >0 was considered as a synergistic event in the DrugCombDB data, whereas second-best synergy ratio <0 in our cohort). Spearman's correlation was computed between the two ranked lists.

### Statistics and reproducibility

We used statistical methods like Wilcoxon rank-sum test[91] to analyze differences in synergy or viability between two groups of cell lines. Multiple hypothesis correction (FDR correction) was done. Fisher exact test was used to test for enrichment. The code and data for main analysis has been made available for reproducibility. No data was excluded from the analyses. No statistical method was used to predetermine sample size. The experiments were not randomized. The investigators were not blinded to allocation during experiments and outcome assessment.

### Reporting summary

Further information on research design is available in the Nature Portfolio Reporting Summary linked to this article.

## Data availability

The processed drug combination screen data are provided in Supplementary Data S1, S2, S8. The raw original screen data are available in Supplementary Data S12. Protein–protein interaction network scores from the publicly available STRING database[71] (downloaded on Aug. 8, 2019) were used.

## Code availability

R software was used the analysis. Code and data for the key analysis of this work is provided here: https://github.com/nishanth83/NSCLC.git

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

## Acknowledgements

This research was supported in part by a grant from the Wellcome Trust (102696) and by the Intramural Research Program of the NIH, National Cancer Institute, and the Center for Cancer Research. This work used the computational resources of the NIH HPC Biowulf cluster (http://hpc.nih.gov). The results shown here are in part based upon data generated by the TCGA Research Network: https://www.cancer.gov/tcga. We thank Xeni Mitropoulos and Dr. Sanju Sinha for their help on this manuscript.

## Author contributions

Conception and design: A.A.F., A..A., M.J.G., J.A.E., D.A.H., C.J.T., E.R., C.H.B. Data collection: X.Z., E.C., A.D., R.K.E., E.M., M.C., G.S.C., J.M., C.K-T., J.L.B., L.J.D., J.H., A.T., C.M., S.M., Z.I. Data Analysis: N.U.N., P.G., A.D.S., J.S.L., E.B., K.M.W., Manuscript preparation: N.U.N., E.R., C.H.B.

## Competing interests

E.R., is a non-paid scientific consultant and cofounder (divested) of Pangea Therapeutics (www.pangeamedicine.com), which focuses on synthetic lethality based precision; he is also a co-founder of MedAware and Metabomed Ltd. C.H.B., is employee of Novartis and previously received research funding from Novartis. A.F., is an employee at Scorpion Therapeutics. A.D.S. provides consultancy to Lead Pharma, Checkmate Pharmaceuticals and C-Reveal Therapeutics. D.A.H., is cofounder of Tell-Bio and on the SAB of Rome Therapeutics. All other authors declare no conflict of interest.
