## [Peer Review File · Nature Communications]

A landscape of response to drug combinations in
non-small cell lung cancerThis manuscript has been previously reviewed at another journal that is not operating a transparent peer review scheme. This document only contains reviewer comments and rebuttal letters for versions considered at *Nature Communications*.

REVIEWERS' COMMENTS

Reviewer #1 (Remarks to the Author):

Overall, the paper is much improved in this revision. The authors have added a number of important validation screens and analyses to confirm their results. The extended discussion section also adds valuable caveats to the findings, discusses the pitfalls of their screen, addresses the challenges of reproducibility in cell line screens, and generally provides a much more nuanced view of their data.

I have only two minor critiques:

- I am a little unclear on the rationale for using a cell line isolated from a cockerspaniel's kidney in the 1950s as a control for human lung tissue. The HUVEC cells are at least derived from human (embryonic) tissue.

- After comparison to the control lines, the combinations were narrowed down to 9 hits. I wonder if these would stand up to statistical significance. That is, if you take the test statistics for all of the original 20+ hits then run Benjamini-Hochberg at some reasonable alpha (e.g. 10%), how many truly significant discoveries are left?

In summary, this paper is now in good shape. It makes a substantial contribution to the combination drug screening literature. The revisions make it above the bar for publication and the resource provided will be useful for downstream analyses.

Reviewer #2 (Remarks to the Author):

The authors have addressed all my remaining concerns. I appreciate the new work, and honest presentation, on the primary and validation screen data concordance.

Reviewer #4 (Remarks to the Author):

The authors have answered many comments raised by the original reviewers in their revised manuscript. The addition of further data significantly strengthens the manuscript. Like this drug combination study database will provide very fruitful information to the research community after the data is available on the web.

I do have only minor comments on the current revised manuscript.

Minor comments

(1) In fig1. Several characters become garbled characters: (Foretinib, Crizotinib, etc...)

(2) The position of Fig 1A would be better on the top, left. Please arrange the position of each figure in Fig 1.

(3) Fig 2B and 2D, isn't it possible to write cell line name on the top? In addition, isn't it difficult to change a color (darkness and lightness) based on the HSA score or synergy score?

(4) Fig 3G, BMS-754807 is IGF1R inhibitor but not IGF11R.

(5) The legend of Fig 5 is not kind for the readers.

(6) P40, line 12, We did his.... -> We did this

RESPONSE TO REVIEWERS' COMMENTS

Thank you for your very helpful reviews. The reviewers' comments are shown in blue. Please find our responses in black. The changes we made to address them (in the manuscript or supplementary material) are shown in red.

Reviewer #1 (Remarks to the Author):

Overall, the paper is much improved in this revision. The authors have added a number of important validation screens and analyses to confirm their results. The extended discussion section also adds valuable caveats to the findings, discusses the pitfalls of their screen, addresses the challenges of reproducibility in cell line screens, and generally provides a much more nuanced view of their data.

Response: Thank you very much for appreciating our work.

I have only two minor critiques:

- I am a little unclear on the rationale for using a cell line isolated from a cockerspaniel's kidney in the 1950s as a control for human lung tissue. The HUVEC cells are at least derived from human (embryonic) tissue.

Response:

This is a control for general toxicity on mammalian cells. This cell line has been used in many studies to assess potential toxicity on mammalian cells in various viability assays as well as drug absorption. Thus, there is a large reference dataset allowing to compare the toxicity of our treatments to those previously studied for cancer treatment.

- After comparison to the control lines, the combinations were narrowed down to 9 hits. I wonder if these would stand up to statistical significance. That is, if you take the test statistics for all of

the original 20+ hits then run Benjamini-Hochberg at some reasonable alpha (e.g. 10%), how many truly significant discoveries are left?

Response: Thank you very much for your suggestion. We earlier did not do a statistical test as we had only 2 normal non-cancerous cell lines as opposed to 15 cancerous cell lines. However, following the reviewer's comments, we used the replicate data to carry out statistical significance tests (there were 4 replicates per drug combination in each cell line) between the cancer and normal cell lines. We found that among the 9 drug combinations that we identified earlier, 8 of them were more effective in cancer cell lines in comparison to the non-cancerous cell lines in a statistically significant manner (Benjamini-Hochberg or FDR < 0.1). The main text has been modified accordingly, as follows:

In the main manuscript (pages 23, 24):

“Furthermore, to study how combinations might be broadly toxic across cellular lineages rather than selectively effective against NSCLC cells, we also carried out drug combination experiments on two non-cancerous immortalized cell lines (HUVEC and MDCK; Table S8). We, therefore, tested the 29 drug combinations that we have studied above in 15 NSCLC cancer cell lines. **Out of those**, we identified 9 drug combinations that are more effective (less viability) in at least 80% ($n \geq 12$) of the cancer cell lines than they are in the two immortalized cell lines (Methods; Table S11A); **8 out of them were found to be statistically significant (we analyzed the replicate data to carry out statistical significance tests (there were 4 replicates per drug combination in each cell line), using a one-sided Wilcoxon rank-sum test at FDR < 0.1).** Repeating the analysis using synergy scores (instead of viability) showed that there are 11 drug combinations that are more synergistic in cancer compared to the immortalized cell lines (Methods; Table S11B; **10 out of them were found to be statistically significant, one-sided Wilcoxon rank-sum test at FDR < 0.1).** **Notably, three** drug combinations were identified in both the viability and synergy-based screens, including Adavosertib and AZD7762, Nilotinib and Trametinib, and Imatinib and Trametinib.”

In the Methods section (page 44) we wrote:

“For computing statistical significance of either more effective or synergistic drug combinations in cancer cell lines in comparison to non-cancerous cell lines, we considered all the 4 replicates (block IDs) and computed one-sided Wilcoxon rank-sum test between viability or synergy scores between cancer and normal cell lines (FDR < 0.1).”

In summary, this paper is now in good shape. It makes a substantial contribution to the combination drug screening literature. The revisions make it above the bar for publication and the resource provided will be useful for downstream analyses.

Response: Thank you very much for your positive feedback.

Reviewer #2 (Remarks to the Author):

The authors have addressed all my remaining concerns. I appreciate the new work, and honest presentation, on the primary and validation screen data concordance.

Response: Thank you very much for appreciating our work.

Reviewer #4 (Remarks to the Author):

The authors have answered many comments raised by the original reviewers in their revised manuscript. The addition of further data significantly strengthens the manuscript. Like this drug combination study database will provide very fruitful information to the research community after the data is available on the web.

Response: Thank you very much for your appreciative comments.

I do have only minor comments on the current revised manuscript.

Minor comments

(1) In fig1. Several characters become garbled characters: (Foretinib, Crizotinib, etc...)

Response: Thanks. We have addressed this.

(2) The position of Fig 1A would be better on the top, left. Please arrange the position of each figure in Fig 1.

Response: Thanks.

The arrangement of the panels was chosen to optimize display and space to allow for larger size of complex panels. Guidance of the editors is requested to as to whether the order of the panels should be changed.

(3) Fig 2B and 2D, isn't it possible to write cell line name on the top? In addition, isn't it difficult to change a color (darkness and lightness) based on the HSA score or synergy score?

Response: Thanks.

The size of the display does not allow for the cell line names to be listed without becoming illegible. We have tried quite a few options there before settling on the current display.

The point of the figure is to allow an easy enumeration of the number of synergistic events by the reader. Creating color scale that would apply to all the panel doesn't improve the outcome at least in our trials. We have spent a considerable amount of time trying to optimize the various display of the results of this very large screen.

(4) Fig 3G, BMS-754807 is IGF1R inhibitor but not IGF11R.

Response: Thanks.

Thank you for catching this typo: The figure has been corrected.

(5) The legend of Fig 5 is not kind for the readers.

Response: Thanks. This has been addressed as follows in Figure 5 legend:

“Figure 5: Summary of the validation screen results.

Eight anchor drugs from the original screen were combined with a specific set of library drugs as indicated. The synergy score (Bliss score, green gradient, left section as labeled on the top) and HSA score (Orange gradient, right section as labelled on the top) is shown for each combination tested. Each column of colored squares corresponds to one of the 15 cell lines tested.”

(6) P40, line 12, We did his.... -> We did this

Response: Thanks for point this out. We have corrected this.